# Mean-Field RL for Large-Scale Unit-Capacity Pickup-and-Delivery Problems

**Kai Cui**[12]**, Sharif Azem**[1]**, Christian Fabian**[1]**, Kirill Kuroptev**[1]**, Ramin Khalili**[2]**, Osama Abboud**[2]**,
Florian Steinke**[1]**, Heinz Koeppl**[1]
[1] *Technische Universität Darmstadt,* [2] *Huawei*
{kai.cui.de, sharifazem1}@gmail.com, {christian.fabian, kirill.kuroptev}@tu-darmstadt.de,
{ramin.khalili, osama.abboud}@huawei.com, {florian.steinke, heinz.koeppl}@tu-darmstadt.de

*Reviewed on OpenReview:* *https://openreview.net/forum?id=E8JRswdyDR*

## Abstract

Solving large-scale vehicle routing problems (VRPs) is NP-hard and poses a computational challenge in numerous applications such as logistics. Meanwhile, mean-field control (MFC) provides a tractable and rigorous approach to controlling many agents. We provide a solution to pickup-and-delivery VRPs via scalable MFC. In combination with reinforcement learning (RL) and clustering, our MFC approach efficiently scales to large-scale VRPs. We perform a theoretical analysis of our MFC-based approximation, giving convergence results for large VRP instances and error bounds for clustering-based approximations. We verify our algorithms on different datasets and compare them against solutions such as OR-Tools, PyVRP and heuristics, showing scalability in terms of speed for mean-field methods, for the first time in discrete optimization. Overall, our work establishes a novel synthesis of MFC-based RL techniques, vehicle routing problems and clustering approximations, to solve a hard discrete optimization problem of practical use in a scalable way.

## 1 Introduction

Solving large-scale vehicle routing problems (VRPs) is an area of practical interest in operations research. There are numerous applications, ranging from dial-a-ride (Cordeau & Laporte, 2007; Ho et al., 2018) over shared use of autonomous vehicles (Narayanan et al., 2020; Lin et al., 2012), or delivery logistics (Bortfeldt & Yi, 2020; Wang et al., 2021; Arunapuram et al., 2003; Zhang & Lygeros, 2023) to agriculture (Bochtis & Sørensen, 2009; 2010). In this work, we focus on a specific variant: the unit-capacity pickup-and-delivery problem (PDP), where a large fleet of vehicles must serve an equally large number of transport requests. The unit-capacity case can also be solved as a capacitated VRP (CVRP), and beyond unit-capacity our framework also applies, but scalability challenges remain. This problem structure is motivated by emerging large-scale logistics challenges with thousands of agents, such as in ride-sharing, food delivery and last-mile gig-economy services (e.g., Amazon Flex). Due to the immense number of vehicles and customers, scalable solutions remain an active area of research (Konstantakopoulos et al., 2022; Zhang et al., 2022).

Motivated by complex applications with many vehicles and tasks, there has been growing interest in solving large-scale VRPs via machine learning. Learning approaches for finding scalable VRP solutions are manifold and differ in their conceptual design. Some learning techniques split the initial VRP into smaller and more tractable subproblems and delegate their solution to subsolvers (Li et al., 2021b) to reduce the overall computational complexity. Other research (Lei et al., 2022) uses transformers to increase the scalability of learning algorithms. For recent overviews on learning VRPs, see Li et al. (2022); Bogyrbayeva et al. (2024).

### 1.1 Contribution

Scaling solutions for very large NP-hard problems can be difficult. For this reason, our approach is based on mean-field control (MFC) (Fornasier & Solombrino, 2014; Bensoussan et al., 2013) as the cooperative form of

mean-field games (Lasry & Lions, 2007; Saldi et al., 2018). This will allow our method to scale to arbitrarily large problem sizes, with constant problem complexity in the limiting problem, and linear complexity in the number of agents / objectives for the actual finite problem, by sampling agent routes. The idea of MFC is to abstract a large number of indistinguishable agents – in our case vehicles – into their *mean-field* (MF) probability distribution. Working with the MF distribution entails both solid theoretical properties as well as reduced computational complexity which, in turn, paves the way for scalable learning algorithms. In contrast to existing works, our MF-based solution allows to solve arbitrarily large problems at constant computational complexity. We propose MFC-based reinforcement learning (RL) for solving large-scale VRPs with pickup and delivery, where our learning approach scales to arbitrarily large numbers of vehicles and deliveries.

Our contributions are summarized as: (i) We present a first formulation of vehicle routing – a hard discrete optimization problem – as MFC & RL; (ii) We give general and special fine-tuned algorithms for solving large-scale vehicle routing problems, considering both general pretraining as well as fine-tuning to problems; (iii) We prove convergence to the MF limit for the case of large-scale VRPs with many vehicles and locations. Moreover, we show approximate probable optimality of MFC solutions and give error bounds for general metric spaces with clustering approximations; (iv) We empirically evaluate by comparing against OR-Tools, PyVRP and heuristics, showing potential generalization capabilities and speeds of MFC for different large-scale settings, where existing methods for discrete optimization can be expensive.

## 1.2 Related Work

In comparison to our work, a majority of recent related work does not consider pickup-and-delivery problems. We also find that few other RL works allow few minute fine-tuning times for large instances, a speed our method achieves. Moreover, very few works consider large-scale VRPs with tens of thousands of nodes and vehicles. In the following, we discuss those works that are most related to our work.

**VRP.** For example, Zhou et al. (2024) consider different variants of VRPs, but no pickup-and-delivery, focusing on multi-task settings with generalization and presented for up to 100 nodes. Similar smaller settings are found in the following works, such as DPN (Zheng et al., 2024), which considers min-max VRPs via sequential RL-based partition and navigation methods. In contrast, we consider near-arbitrarily large VRP. In Liu et al. (2024), VRP with heterogeneous capacities are considered via RL-based solutions, and evaluated for small fleet and customer sizes of up to 7 and 100. As for other recent work, Wu et al. (2024) study multi-objective VRPs using collaborative RL, Xiao et al. (2024) demonstrate fast transformer-based solutions, and Kikuta et al. (2024) consider smaller variants of VRP for electric vehicles. On the other hand, Min et al. (2024) propose graph-learning methods for traveling salesman problems (TSP), and Jiang et al. (2023) focus on out-of-distribution generalization. Furthermore, Xia et al. (2024) explore heuristics for the special case of TSPs, highlighting scalability issues with current solutions and proposing scalable Monte Carlo tree search. Our contribution similarly is a method that provides an automatically scaling solution for certain VRPs.

**Large-scale combinatorial optimization.** As for literature on large-scale combinatorial optimization, recent works include meta-learning and reinforcement learning for combinatorial optimization (Qiu et al., 2022), graph-based diffusion solvers (Sun & Yang, 2023), and methods that generalize pre-trained models or learned heuristics to arbitrarily large TSP or VRP instances (Fu et al., 2021; Hou et al., 2023). Other neural combinatorial optimization approaches also aim for large-scale generalization via encoder-decoder architectures (Luo et al., 2023). In contrast, our MFC-based RL method provides a methodology for reducing multi-agent RL (MARL) to *single-agent* RL of *constant complexity*, regardless of number of agents.

**Pickup-and-delivery problems.** Lastly, most related to our setting, also known as pickup-and-delivery problems (PDP), Li et al. (2021a) and Zong et al. (2022) solve single-agent and multi-agent PDP using deep attention-based (multi-agent) reinforcement learning methods. A direct comparison with these methods can be challenging due to the difference in scale. For instance, the MAPDP framework (Zong et al., 2022) uses a Transformer that computes attention across all nodes, leading to a computational complexity that scales quadratically with the number of nodes. This eventually becomes intractable for large problems such as the 50,000-node problems that are a primary focus of our work.

While it is true that recent learning-based routing solvers with heavy decoders, such as BQ-NCO (Drakulic et al., 2023), LEHD (Luo et al., 2023), and SIGD (Pirnay & Grimm, 2024), in fact demonstrate generalization

to large-scale CVRP instances (e.g., up to 100k nodes by applying subquadratic attention techniques (Luo et al., 2025)), applying them directly to our problem is non-trivial. First, as discussed, our unit-capacity PDP can only be reformulated as a CVRP with asymmetric non-Euclidean distances. Second, some adaptation would be needed to handle the specific pickup-and-delivery structure, while higher than unit capacity cannot be modelled as asymmetric CVRP. To use existing solvers in this manner would likely require a distance-matrix-based encoder, such as those in Kwon et al. (2021) or Son et al. (2025), rather than a node-coordinate-based one. Given these required adaptations, we consider a direct benchmark to be beyond the scope of this foundational paper, which instead introduces a new paradigm for extreme-scale PDP.

**Mean-field modelling.** Modelling agent MFs is done in analogy to general MFC (Angiuli et al., 2022; Carmona et al., 2023; Cui et al., 2021), with policy gradients (Angiuli et al., 2023; Carmona et al., 2019; Cui et al., 2024) as a solution for the resulting high-dimensional RL problem. However, VRPs require a generalized formulation with *delivery* distributions: We first suitably rewrite between VRP and multi-agent control with MFs. Moreover, we have two coupled MFs not modelled by general MFC-based MARL, as the two MFs of agents & objects create a double limit, which we resolve by introducing a ratio of objects and agents. To our knowledge, our work is the first to consider MF-based combinatorial optimization such as VRP, in contrast to related but distinct traffic flow analysis using mean-field games Huang et al. (2021). Importantly, we also obtain probable approximate optimality of any realization in MFVRP, not obtained in standard MFC, via embedding finite MFVRP into the limiting system, see Section 2.5 for a discussion.

## 2 Mathematical Framework

In this section, we connect VRP, MARL and MFC. Proofs for theoretical results are found in the Appendix.

*Notation: For any integers $k, l$, let $[k] := \{1, 2, \ldots, k\}$ and $[k] + l := \{l + 1, \ldots, l + k\}$. For any metric space $A$, let $\mathcal{B}_1(A)$ denote Borel measures bounded by 1 over $A$, with probability measures $\mathcal{P}(A) \subseteq \mathcal{B}_1(A)$.*

### 2.1 Capacitated VRP with Pickup and Delivery

The finite problem of interest is a vehicle routing problem with pickup and delivery. We consider a large number of $N$ vehicles and $M$ objects, each of which needs to be transported from one position to another in some metric space $\mathcal{X}$. The methodology applies to any metric space. For example, $\mathcal{X}$ can be a subset of Euclidean space, or it could be nodes on a road graph with appropriate node distances. We index vehicles and objects by indices in $\mathcal{N} := [N]$ and $\mathcal{M} := [M] + N$ respectively. We denote distances between objects or vehicles at $x, y \in \mathcal{X}$ as $d(x, y)$. For example, we consider Euclidean spaces $\mathcal{X} \subseteq \mathbb{R}^2$ with the Euclidean distance $d(x, y) = \|x - y\|_2$ as well as finite spaces embedded into the Euclidean space. More generally, we also include positions on a road network or graph. Each object or transport mission $j \in \mathcal{M}$ is specified by its source $s_j \in \mathcal{X}$ and destination $d_j \in \mathcal{X}$, while each vehicle $i \in \mathcal{N}$ starts at an initial depot $s_i \in \mathcal{X}$.

For simplicity of exposition, consider vehicles with carrying capacity of one object. A more general setting with larger capacities can be considered analogously, see also Section 2.5, though the state-action complexity will be exponential in the transport capacity. The goal is to optimize *which* vehicles transport *which* objects in *which* order, such that an overall global transport cost is minimized. Formally, the overall cost is given as the total distances travelled. That is, we define the set of all indices $\mathcal{V} := [N + M] = \mathcal{N} \cup \mathcal{M}$ and optimize the *objective* $J := \sum_{k \in \mathcal{N}} \sum_{i,j \in \mathcal{V}} d_{ij} b_{ij}^k$ for binary decision variables $\mathbf{b} := (b_{ij}^k)_{a \in \mathcal{N}, i,j \in \mathcal{V}}$, where $b_{ij}^k$ is one whenever vehicle $k$ moves from depot or transport mission $i$ to $j$ with induced travel cost between depots or mission sources $i, j \in \mathcal{V}$

$$d_{ij} := d(s_i, d_i) + d(d_i, s_j), \tag{1}$$

where the depot's source and destination are equal to the depot's position.

We consider a standard tour length constraint $C \in \mathbb{N}$ as maximum number of missions to perform by any single vehicle, e.g., time or energy constraints. At least for carrying capacity of one, the above problem can also be solved via CVRP by considering each mission as one location, but with asymmetric non-Euclidean distances (1) due to differing pickup and delivery locations. The optimization is well known to be NP-hard. We introduce MF-approximations for the important case of many missions and vehicles.

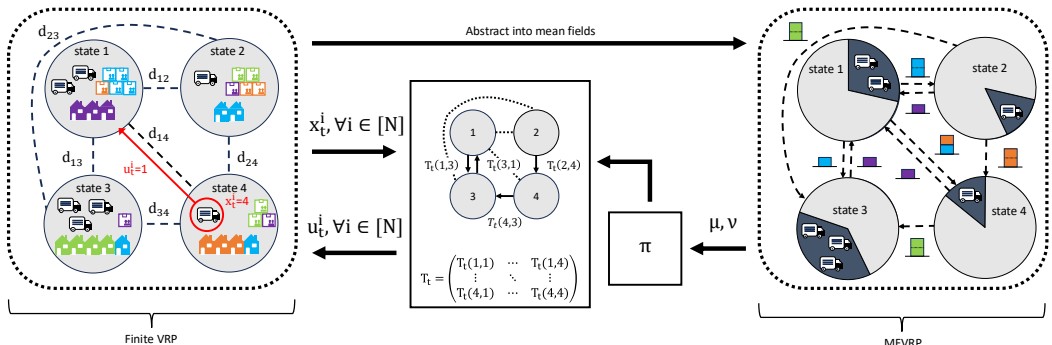

Figure 1: Simplified overview of finite VRP and MFVRP. Exemplary system with four states, seven trucks and 14 packages and destinations. Orange (blue / green / purple) packages should be delivered to orange (blue / green / purple) destinations / houses. For Euclidean locations, each location is assigned to one state via clustering, see Section 3.

## 2.2 VRP as Mean-Field Multi-Agent Control

We consider a sequential solution of VRPs formulated as a multi-agent control problem, i.e. performing one delivery per vehicle in each "time step" $t \in \mathbb{N}$ (decision epoch, not real time). The multi-agent control formulation of VRP is obtained by assuming an underlying distribution of vehicle and object locations, and introducing both current vehicle states and remaining transportation missions. That is, we assume each location $x_0^i$ of vehicle $i \in \mathcal{N}$ – the vehicle state – is sampled from some initial distribution $\mu_0 \in \mathcal{P}(\mathcal{X})$, $x_0^i \sim \mu_0$. Analogously, the source-destination tuples $(s_j, d_j)$ of objects $j \in [M]$ are sampled from an object distribution $\nu_0 \in \mathcal{P}(\mathcal{X}^2)$, $(s_j, d_j) \sim \nu_0$. Consider a fixed ratio $R = M/N$ of objects per vehicle to transport. Then, due to indiscernibility of objects and vehicles at the same locations, the entire information of a VRP is contained in the parameter $R$ and the empirical distribution of vehicle states $\mu_0^N = \frac{1}{N} \sum_i \delta_{x_0^i}$ and source and destination locations $\nu_0^N = \frac{1}{M} \sum_j \delta_{(s_j, d_j)}$ – the empirical MFs at time zero. Here, the MFs can be understood as "histograms" over $\mathcal{X}$ and $\mathcal{X}^2$ respectively, with Dirac measures $\delta$. For a visualization, see the left side of Fig. 1, which is the original VRP and is transformed into a problem of vehicle and object flows.

The current intermediate state of a VRP solution during execution at any time $t$ is given by counting the remaining objects to be transported at each source and destination, as well as the current positions of vehicles. Indeed, the current state is contained in $\nu_t^N = \frac{1}{M} \sum_{j \text{ not transported yet}} \delta_{(s_j, d_j)}$ and $\mu_t^N = \frac{1}{N} \sum_i \delta_{x_t^i}$. Here, the states of vehicles $x_t^i$ are moved by vehicle actions $u_t^i \in \mathcal{X}$ to $x_{t+1}^i = u_t^i$. When moving, each vehicle $i$ transports a random object from $x_t^i$ to $u_t^i$, if one exists.

The reward is then given by the negative distance travelled by all vehicles plus a reward for each delivered package, $r_t^N = c_\mathrm{d} \cdot \#_t - \sum_i d(x_t^i, u_t^i)$, where $\#_t$ is the number of delivered objects in epoch $t$ and constant $c_\mathrm{d} \geq 2 \operatorname{diam}(\mathcal{X})$. The choice of $c_\mathrm{d}$ ensures that an optimal undiscounted control maximizing $J^N = \mathbb{E}\left[\sum_t r_t^N\right]$ is also an optimal VRP solution: the reward for a single delivery must outweigh the maximum possible travel distance saved by skipping it. Analogously, we add a final cost for moving back to the depot. We award a larger final reward $r_\mathrm{bonus}$ whenever all objects have been transported; this is necessary to properly incentivize completion, as agents are also penalized for the final return trip to the depot. A cost of $c_\mathrm{miss}$ times the fraction of objects not delivered is added as a reward shaping term to guide the RL agent, though it is not strictly necessary for optimality. Here, we terminate an episode when all objects are delivered or after reaching an episode time horizon of $t_\mathrm{max} = C$ to respect the maximum tour length constraint $C \in \mathbb{N}$.

## 2.3 Mean-Field VRP

To obtain tractable MFVRP, in the infinite vehicle-object limit $N \to \infty$ with $M = RN$, $R > 0$ the initial empirical MFs are given by $\mu_0$ and $\nu_0$ via law of large numbers. It suffices to describe the system by limiting deterministic MFs $\mu_t \in \mathcal{P}(\mathcal{X})$, $\nu_t \in \mathcal{B}_1(\mathcal{X})$ at all times $t$. Here, at each time $t$, the MFVRP action is a

transportation by each of infinitely many vehicles, i.e. we assign all vehicles from current to next location and transport objects on the way. In general, the action at time $t$ is therefore a transport map $T_t$ on $\mathcal{X}$.

For finite $\mathcal{X}$, the action can be numerically represented as a transition matrix $T_t \colon \mathcal{X} \times \mathcal{X} \to [0,1]$ with $\sum_d T_t(s,d) = 1$ for all $s \in \mathcal{X}$, encoding which fraction $T_t(s,d)$ of vehicles at $s \in \mathcal{X}$ will move to $d \in \mathcal{X}$. Since the vehicle-object-ratio is $R > 0$, the remaining transportation tasks decrease according to

$$\nu_{t+1}(s,d) = \max(0, \nu_t(s,d) - \mu_t(s)T_t(s,d)/R) \tag{2}$$

while the MF of vehicles evolves as

$$\mu_{t+1}(d) = \sum_{s \in \mathcal{X}} \mu_t(s)T_t(s,d). \tag{3}$$

Hence, the MFC problem is a single-agent Markov decision problem, with states $(\mu_t, \nu_t)$ and actions $T_t$, where the reward is given by delivered objects minus distance travelled (on average per agent, which is equivalent to optimizing the total travelled distance)

$$r_t = \sum_{s,d \in \mathcal{X}} c_{\mathrm{d}} \min(\nu_t(s,d), \mu_t(s)T_t(s,d)/R) - d(s,d)\mu_t(s)T_t(s,d). \tag{4}$$

In other words, we learn an "upper-level" MFC policy $\hat{\pi}$ via RL, such that $T_t \sim \hat{\pi}(T_t \mid \mu_t, \nu_t)$. See also Fig. 1 for a visualization. Final rewards and costs are handled analogously. We choose $c_{\mathrm{d}} > 2\operatorname{diam}(\mathcal{X})$ to ensure optimality of delivering all objects instead of waiting until termination.

The dynamics of our MFVRP formulation exhibit a strong connection to Optimal Transport (OT). Equation (3) is a transport map that pushes the vehicle distribution $\mu_t$ forward to $\mu_{t+1}$. Equation (2) resembles a continuity equation with a sink term, where the mass of the object distribution $\nu_t$ is depleted by the flow of vehicles. Our problem can be interpreted as finding a sequence of conditional transport plans $\{T_t\}$ that steer the coupled distributions $(\mu_t, \nu_t)$ to minimize total transport cost, subject to the constraint of depleting $\nu_t$. This perspective links our work to dynamic OT formulations Santambrogio (2010), though our setting is distinguished by the coupled discrete-time dynamics between optimal transport flow $\mu_t$ and object flow $\nu_t$.

### 2.4 Theoretical Guarantees

To show optimality of MFC in VRP, the three introduced models are rigorously analyzed. We first sketch how the VRP objective is exactly given by the finite MARL system objective, and then how the MARL objective is approximately solved by the limiting MFVRP MDP.

**MARL vs. VRP.** The connection is made by noticing that vehicle trajectories in the MARL system $(x_0^i, u_0^i, x_1^i, u_1^i, x_2^i, \ldots)_{i \in \mathcal{N}}$ correspond to a solution of the VRP. Any objects not transported by the trajectory are handled by postprocessing, by transporting objects by random available vehicles at the end. Assuming optimal behavior of vehicles in the MARL problem and therefore that all objects are transported, the cost of the VRP as total distance travelled is exactly given by the negative sum of rewards $J$, up to a constant episodic reward for delivering all objects. Therefore, optimally solving the MARL problem optimally solves the VRP. In other words, for any vehicle $i \in \mathcal{N}$, we assign $b_{jk}^i = 1$ whenever it transports objects $j$ and then $k$, as well as $b_{ij}^i = 1$ for moving from depot $i$ to the first object $j$ and $b_{ji}^i = 1$ for the last object, thereby removing any unnecessary in-between movements of each vehicle that does not transport objects.

**MARL vs. MFC.** A MFC solution solves the infinite-vehicle model, and approximates optimal solutions in the finite MARL system by allowing the sampling from $T_t$ – a random realization of the deterministic MFC system (2) and (3) constitutes a MARL solution and therefore also VRP solution. Vehicles take actions according to deterministic, or hence equivalently constant, $\pi_t \equiv T_t$ at all times $t \leq t_{\max}$ up to termination at time $t_{\max}$, after which we randomly transport any remaining objects. For a visualization see Fig. 1.

An error analysis of finite VRP gives us estimates of the asymptotic error of MFC assuming an infinitude of vehicles and objects, instead of finitely many. We then learn a deterministic MFC policy $\hat{\pi}$, e.g., after the

convergence of stochastic policy gradient methods (Schulman et al., 2017), or by using deterministic policy gradients (Silver et al., 2014), which can be provably near-optimal in large finite problems.

**Theorem 2.1** (Propagation of Chaos)**.** *Let $\mathcal{X}$ be finite and consider some deterministic $\hat{\pi}$. Then, at any time $t \in \mathbb{N}$, the error between the finite-agent empirical measures and the mean-field measures is $\mathcal{O}(1/\sqrt{N})$, i.e.*

$$\mathbb{E}\left[\|\mu_t^N - \mu_t\|_\infty\right] = \mathcal{O}(1/\sqrt{N}) \quad and \quad \mathbb{E}\left[\|\nu_t^N - \nu_t\|_\infty\right] = \mathcal{O}(1/\sqrt{N}).$$

*Consequently, $\|\mu_t^N - \mu_t\|_\infty \to 0$ and $\|\nu_t^N - \nu_t\|_\infty \to 0$ in probability as $N \to \infty$.*

As a result, we can obtain approximate optimality bounds given a finite $N$-agent VRP.

**Corollary 2.2** (Probable Approximate Optimality)**.** *Let $\mathcal{X}$ be finite. An optimal MFC solution is probably approximately optimal in the finite $N$-agent system. Specifically, for any failure probability $\delta \in (0, 1)$, there exists a constant $\mathcal{C}_1$ independent of $N$ and $\delta$ such that for any $N \in \mathbb{N}$, the optimality gap is bounded by $\frac{\mathcal{C}_1}{\delta\sqrt{N}}$ with probability at least $1 - \delta$.*

Our theoretical results show that MFVRP is approximately optimal in large VRP. Moreover, the results are PAC-style guarantees which would give us exact probably-approximate optimality bounds for any fixed finite $N$-agent VRP instance in practice. The concrete constants can be computed inductively as in the proofs.

### 2.5 Generality and Limitations

The optimal MFC solution in Corollary 2.2 requires no additional assumptions because we may replace it by an optimal *constant* MFC policy, leading to deterministic $T_t$. This works only for our MFVRP setting and not for general MFC, since we show that any optimal VRP objective is also obtained exactly in the limiting MFC by a suitable MFC policy, see the third inequality in the proof of Corollary 2.2. The advantage of it is constant RL problem size regardless of number of agents / clients around each cluster.

**Generality.** Even for unit-capacity, MFC-based RL is well-justified by virtue of scaling to arbitrarily many agents. The sequential formulation can allow dynamic variants of VRP which are not addressed by standard VRP solvers, e.g., with random arrival of new missions, or stochastic failure of vehicles. MFVRP can further model heterogeneous vehicles, such as heterogeneous fleets with different capacities or speeds, by duplicating the state space for each separate type of vehicle. The methodology also applies to any metric space including non-Euclidean such as on graphs, in practice by using generalized clustering methods. Finally, in case of uncountable $\mathcal{X}$, such as a convex subset of $\mathbb{R}^2$, transport maps $T_t$ are replaceable by couplings on $\mathcal{X} \times \mathcal{X}$, but the resulting MFVRP MDP is of infinite dimension. Instead, for algorithmic tractability, in the following we consider a discretization-based approach, with guarantees of near optimality for sufficiently fine discretization.

**Limitations.** We note that MFC is not without limitations for solving discrete optimization problems, which are discussed in the following. In particular, MFC remains hard to apply for non-pickup-and-delivery VRPs: First, the special case of multiple traveling salesmen problem (mTSP) under MFC would boil down to a clustered version of standard mTSP, since distances between points in the same cluster are zero so they are immediately served by a single same agent. While this case can be considered, there is therefore no methodological novelty in applying MFC to mTSP beyond clustering all the locations and replacing them with their centroid. Meanwhile, constant finite capacities without pickup-and-delivery structure disappear in the limit of infinite agents, and are therefore also difficult to model. For scaling capacities, we obtain back a standard VRP with finite number of agents and not the RL problem we discuss. Further, time constraints are hard to model for MFC.

**Beyond Unit Capacity.** Lastly, a primary limitation of the current formulation is the focus on unit-capacity vehicles. While this simplifies the state representation, it does not directly model many real-world scenarios. However, we can generalize the system to higher-than-unit-capacity problems by exponentiating the state space of agents to model picked-up objects, and generalizing transition matrices $T_t$ with additional probabilities of picking up objects with particular destination, i.e. the state space is exponential in the number of objects per vehicle. To generalize to a capacity of $C_{\text{veh}} > 1$, the agent's state space would need to be augmented to track the set of destinations for all carried objects. However, this would lead to a state-action complexity that scales exponentially with $C_{\text{veh}}$, posing a significant challenge for the RL algorithm.

For example, consider $C_{\text{veh}} = 2$. The new state space is $\mathcal{X}' = \mathcal{X} \cup (\mathcal{X} \times \mathcal{X})$ with states $x \in \mathcal{X}$, or $(x, t) \in (\mathcal{X} \times \mathcal{X})$ defining an agent at location $x$ with an object to be transported to $t$. Then, the possible agent actions are:

1. Move to $v \in \mathcal{X}$ and transport as many objects from $x$ to $v$ (including the carried object from $t$);

2. For an agent at $x \in \mathcal{X}$: Move to $v \in \mathcal{X}$ while picking up an extra object for $t$ from $x$; or

3. For an agent at $(x, t) \in (\mathcal{X} \times \mathcal{X})$: Move to $t$ while picking up an object for $v$ from $x$.

Note that the option to pick two objects or model two carried objects is not useful, since agents transporting two objects by triangle inequality must go to one of the destinations for optimality, and lose at least one object at the end of their action. This keeps the persistent state space quadratic and not cubic. The mean-field state is thus a pair of distributions $(\mu_t^0, \mu_t^1)$ over $\mathcal{X}$ and $\mathcal{X} \times \mathcal{X}$ respectively. Let $T_t(\text{next state}|\text{current state})$ be the policy. The evolution of the mean-field is:

$$\mu_{t+1}^0(d) = \sum_{s \in \mathcal{X}} \left( \underbrace{\mu_t^0(s)T_t(d|s)}_{\text{empty moves to } d} + \underbrace{\mu_t^1(s,d)T_t(d|(s,d))}_{\text{delivers at } d} \right)$$

$$\mu_{t+1}^1(d, t_d) = \sum_{s \in \mathcal{X}} \Big( \underbrace{\mu_t^0(s)T_t((d,t_d)|s)}_{\text{empty at } s \text{ picks one for } t_d, \text{ moves to } d} + \underbrace{\mu_t^1(s,t_d)T_t((d,t_d)|(s,t_d))}_{\text{carries one for } t_d \text{ at } s, \text{ moves to } d}$$
$$+ \underbrace{\mu_t^1(s,d)T_t((d,t_d)|(s,d))}_{\text{at } s \text{ with load } d, \text{ picks up for } t_d, \text{ delivers at } d} \Big)$$

with analogously defined resulting transitions for $\nu$ and rewards. The last term in the second equation captures the complex action where a vehicle at $s$ with a load for $d$ picks up a second object for $t_d$ and immediately travels to $d$ to deliver the first, resulting in the new state $(d, t_d)$. As we can see, the dimensionality of the state space $\mathcal{X}'$ is squared, and the problem quickly becomes intractable. Analogous considerations hold for higher capacities.

## 3 Mean-Field Vehicle Routing RL

We present our algorithmic solution, visualized in Fig. 1. The idea is to solve the MFVRP as a single-agent MDP, via single-agent RL. At the same time, the implementation is multi-agent RL if we directly learn on the finite VRPs. Indeed, this gives us two approaches of implementing algorithms for MFVRP. But first, we describe the clustering procedure required for general metric $\mathcal{X}$ with theoretical error analysis.

### 3.1 Discretization and Clustering

For locations on a road network or graph, a finite state space $\mathcal{X}$ suffices and we can avoid infinite-dimensional MFC state-actions, but in general this is not the case. Therefore, one can use Voronoi partitions to trade-off between accuracy of MF approximation and size of MFC problem. This is particularly useful when the objects and vehicles are clustered around a few most relevant locations such as cities, which are not located on a grid. Consider $k \in [K]$ cell centroids $c_k \in \mathcal{X}$. Each centroid induces a cell $C_k \coloneqq \{x \in \mathcal{X} \mid d(x, c_k) \in \min_j d(x, c_j)\} \subseteq \mathcal{X}$ consisting of all points in $\mathcal{X}$ which are closest to that particular centroid out of all centroids. Then, to minimize the error stemming from MFC approximation, we may use the empirical histogram with cells instead of bins, giving us the approximate MF as fractions of objects to be delivered from one cell to another. As a result, we produce an assignment $\rho \colon \mathcal{M} \to [K]^2$ such that each object $i$ is assigned to the centroids closest to its source and destination location, $\rho(i) \in \arg\min_k d(s_i, c_k) \times \arg\min_l d(d_i, c_l)$, where $\times$ denotes the Cartesian product. In practice, for simplicity we use k-means to produce a clustering of all source and destination locations, assigning each location to one of the clusters. In general one can use any clustering method, e.g., instead minimizing the non-squared Euclidean distances via $k$-medoids (Kaufman & Rousseeuw, 2008). See Fig. 2 for a visualization.

We can then bound the maximum error of the MFC approximation by the maximum sizes of each cluster.

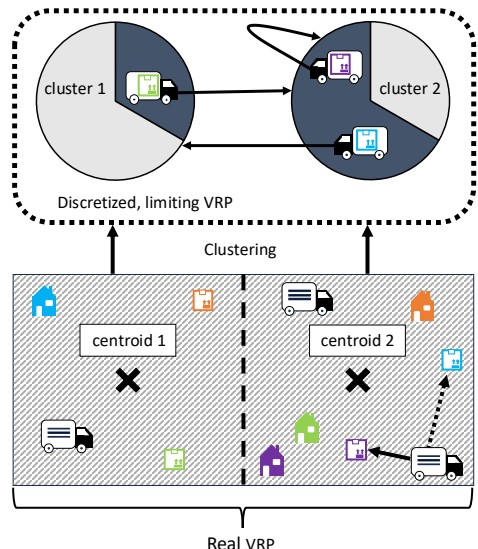

Figure 2: Simplified overview of the clustering approach for Euclidean VRP in a toy example. Observe the correspondence between limiting equations and real VRP, with clustering-based discretization.

**Theorem 3.1.** *An optimal discretized MFC solution is probably approximately optimal in the original (non-discretized) finite $N$-agent system. The optimality gap is composed of a stochastic error from MF approximation as in Corollary 2.2 and a deterministic clustering error. Specifically, for any failure probability $\delta \in (0,1)$, there exists constant $\mathcal{C}_1$ such that with probability at least $1 - \delta$, the optimality gap is bounded by*

$$\underbrace{\frac{\mathcal{C}_1}{\delta\sqrt{N}}}_{Stochastic\ Error} + \underbrace{(4M + 2N) \cdot \max_k \text{diam}(C_k)}_{Discretization\ Error}.$$

Our methodology is hence near-optimal whenever RL converges to an optimal MFC solution, for enough agents and good discretization. Note that time complexity becomes linear in the number of agents or objects $N, M$, because we only need to count how many are in each bin. In contrast, complexity is shifted to quadratic in the number of clusters $K$. Thus, as shown in the experiments, we achieve high scalability in $N, M$.

### 3.2 Algorithm and Fine-Tuning

With the problem of general representation solved, RL can now solve the otherwise difficult-to-analyze optimization or multi-agent control problem. The straightforward approach is to use the limiting equations (2), (3) and (4) to define an environment, and apply single-agent policy gradients to the problem, see Alg. 1.

---

**Algorithm 1** MFVRP-G

---

1: Input: $\mu_0$, $\nu_0$, distance matrix.
2: **for** iterations $n = 1, 2, \ldots$ **do**
3:      **for** time steps $t = 0, \ldots, B_{\text{len}} - 1$ **do**
4:          Sample MFC action $T_t \sim \hat{\pi}^\theta(T_t \mid \mu_t, \nu_t)$.
5:          Compute reward $r(\mu_t)$ and next MFs $\mu_{t+1}$, $\nu_{t+1}$ as in Eqs. (2), (3), (4), and termination / reset flag
         $d_{t+1} \in \{0, 1\}$.
6:      **end for**
7:      Update $\hat{\pi}^\theta$ on minibatch sampled from data $\{((\mu_t, \nu_t), T_t, r_t, d_{t+1}, (\mu_{t+1}, \nu_{t+1}))\}_{t \geq 0}$.
8: **end for**

---

As discussed in Section 2.4, we obtain a valid VRP solution by sampling MARL actions for each vehicle, and performing a postprocessing step where objects not delivered are appended to the route of some random vehicles. Preprocessing is performed via k-means with negligible run time of less than a second per VRP instance, i.e., we assume the data is already clustered into $K$ clusters. Moreover, the depot is appended as an additional cluster centroid at runtime. Whenever a vehicle transports an object from cluster $i$ to cluster $j$, we randomly iterate through vehicles and assign the nearest not yet transported object from $i$ to $j$, see Fig. 2.

For learning a general solution under fixed $K$, we dynamically generate a wide range of scenarios by sampling initial object distributions $\nu_0$ and cluster centroids from flat Dirichlet and uniform distributions, respectively. This can be understood as pretraining on the space of all possible limiting MFVRP scenarios for a given $K$, since we sample from all possible cluster locations and distributions over start and end locations. Alternatively, consider learning on a large fixed data set, e.g. if real data is available and clustered.

**Datasets Fine-Tuning** Beyond learning a general policy, an improved approach is to fine-tune on a specific, fixed VRP instance, using the general policy as a warm start. We explore two ways to do this:

- **MFVRP-F (Mean-Field Fine-tuning):** We continue training on the limiting MFVRP MDP, but use the fixed initial distributions $(\mu_0, \nu_0)$ derived from the specific VRP instance. This approach is fast as it operates entirely in the low-dimensional mean-field space.

- **MFVRP-FD (Finite-Domain Fine-tuning):** We train directly on the finite $N$-vehicle problem. In each step, we sample actions for all $N$ vehicles from the policy's transport map $T_t$, execute them in the environment, and update the policy based on empirical reward and state changes. This avoids a mean-field approximation error during fine-tuning but is computationally more intensive.

However, we find that the latter gives similar results and takes too much time. We thus mainly focus on the former MFVRP-F method. Overall, the discussed approaches result in Algorithm 2. The advantage of fine-tuning is possibly improved results, at the cost of training time.

The post-processing step for both methods involves assigning any leftover objects to random vehicles. We favor this simple approach because for large $N$, the fraction of undelivered objects is theoretically guaranteed to be small under an optimal policy, diminishing as a consequence of our propagation of chaos result (Theorem 2.4). We also performed an empirical verification in Appendix C.1.

---

**Algorithm 2** MFVRP-F/FD

---

1: Input: $\mu_0^N$, $\nu_0^N$, distance matrix, pretrained MFVRP-G model.
2: **for** iterations $n = 1, 2, \ldots$ **do**
3:     **for** time steps $t = 0, \ldots, B_{\text{len}} - 1$ **do**
4:         Sample MFC action $T_t \sim \hat{\pi}^\theta(T_t \mid \mu_t, \nu_t)$.
5:         (F): Compute reward $r(\mu_t)$ and next MFs $\mu_{t+1}$, $\nu_{t+1}$ and flag $d_{t+1} \in \{0, 1\}$ as in Alg. 1 for specific VRP instance.
6:         **for** vehicle $i = 1, \ldots, N$ **do**
7:             (FD): Sample per-vehicle action $u_t^i \sim T_t(u_t^i \mid x_t^i)$.
8:         **end for**
9:         (FD): Perform actions, observe reward $r_t$, next MF $\mu_{t+1}$, termination flag $d_{t+1} \in \{0, 1\}$
10:     **end for**
11:     Update $\hat{\pi}^\theta$ on minibatch sampled from data $\{((\mu_t, \nu_t), T_t, r_t, d_{t+1}, (\mu_{t+1}, \nu_{t+1}))\}_{t \geq 0}$.
12: **end for**

---

In experiments, we consider datasets "finite", "uniform" and "cities". The former consists of a finite state space $|\mathcal{X}| = K$, with distances from randomly sampled locations on $[-1, 1]^2$. The latter takes the latitude and longitude of the 80 largest German cities and randomly samples $K$ cities, around which source and destination locations are normal distributed. For both of the above, we randomly sample a joint distribution over source and destinations in $[-1, 1]^4$ from a flat Dirichlet. Each object is then sampled from this joint distribution, and in the case of "cities" they are additionally perturbed around the location of the city. Lastly,

for "uniform" we sample source and destination locations uniformly at random from $[-1,1]^2$, which gives us a "worst-case" for clustering and MFVRP.

## 4 Numerical Experiments

In the following, we perform an experimental evaluation of our algorithms on large problem instances. The implementation can be found in the supplementary materials.

### 4.1 Implementation Details

For the implementation, we use JAX with PPO (Schulman et al., 2017) based on PureJaxRL (Lu et al., 2022).[1] We compare against three baselines: the OR-Tools routing problem solver (Perron & Furnon, 2023), the recent state-of-the-art PyVRP solver Wouda et al. (2024), and a greedy heuristic.

- **OR-Tools:** We use the standard routing solver, which automatically selects from a portfolio of advanced heuristics and metaheuristics (e.g., guided local search, simulated annealing).

- **PyVRP:** A state-of-the-art open-source solver based on the powerful HGS algorithm. For both OR-Tools and PyVRP, we model the problem as a CVRP with asymmetric distances (Eq. (1)) multiplied by $10^6$ and rounded.

- **Greedy Heuristic:** This is a heuristic corresponding to the 'PATH_CHEAPEST_ARC' first solution strategy in OR-Tools, where each vehicle iteratively travels to the start of the nearest available mission.

We focus our comparison on these scalable, open-source baselines. Advanced commercial solvers were not included as they are proprietary. Similarly, we did not use standard public PDP benchmarks. While valuable, they typically feature smaller instance sizes (e.g., up to 100 nodes) or include problem aspects like precedence constraints which are beyond the scope of this foundational work, whose primary goal is to demonstrate a new algorithmic paradigm for extreme-scale problems.

Each MFVRP-G model is a single policy trained to generalize across diverse problem instances for a *fixed* number of clusters $K$. MFVRP-F then fine-tunes this general policy on one specific VRP instance. For our training, we used a GPU with 40 GB VRAM and over 19 TFLOPS of FP32 compute performance for a single hour for each fixed $K$ in MFVRP-G (pretraining), and around two to three minutes per MFVRP-F or MFVRP-FD run for a particular problem instance.

We learned policies with two hidden layers of 256 nodes and tanh activations. The discount factor is set to $\gamma = 0.99$, and with GAE $\lambda = 0.95$. The minibatch sizes are 10000 for MFVRP-F, as well as annealed from 10000 to 250000 for MFVRP-G, with 8 PPO steps per training batch. The clip parameter is set to 0.2. The learning rate was linearly annealed from 0.00003 down to zero. As usual in RL, our implementation uses discounted objectives and stationary time-independent policies (Guo et al., 2022), and we also transform the current time $t$ into a one-hot observation. Unless stated otherwise, we use the parameters $K = 5$, $R = 5$, $C = 15$, $c_d = 50$, $c_{\text{miss}} = 30$, $r_{\text{bonus}} = 5$ and datasets for clustered and uniform scenarios with all locations in $\mathcal{X} = [-1,1]^2$, generated as described in the Appendix. We perform experiments on the aforementioned 3 types of datasets using the parameter $K$ with 50 instances each, to be able to assess the performance and generalization of our algorithms in different scenarios. For uniform datasets, in the following $K$ is merely a hyperparameter of the algorithm.

### 4.2 Experimental Results

In Fig. 3, we show the training curves for the general MFVRP-G trained on $K = 3$ and $K = 5$. The curve is smooth as we chose a sufficiently high batch size (10000 to 250000), making gradient estimates less noisy. We observe that the methodology converges in terms of episodic returns, and in turn that the maximization

---

[1]Code can be found under `http://github.com/tudkcui/MFVRP`

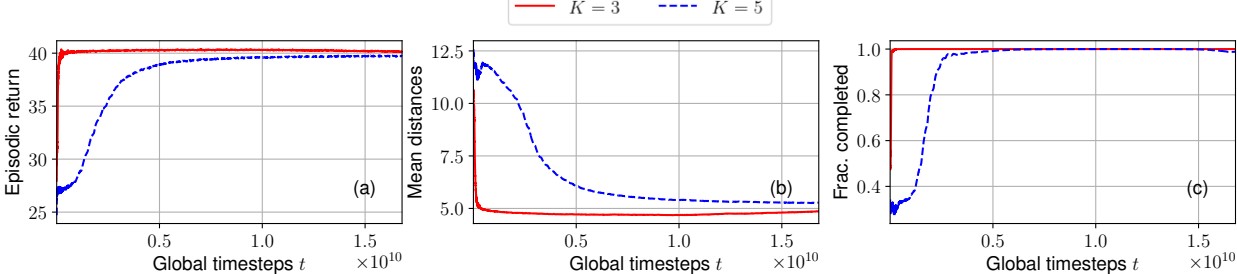

Figure 3: Training curves for MFVRP-G. Comparison of returns, travelled distances, and percentage of completed episodes for MFVRP-G.

of episodic returns indeed minimizes the travelled distances of vehicles while completing all episodes. The results reach a close to 100% completion rate of deliveries while optimizing the distances travelled. This verifies our reward function design.

We can also see that training takes longer as $K$ increases, and uses a great number of time steps, as common for RL methods. In particular, the batch sizes were chosen large to obtain stable training behavior, since the MFVRP MDP has the unique RL challenge of high-dimensional continuous action spaces. Nevertheless, we observe that RL begins to reduce the return and fraction of completed episodes when training for too long, and we apply early stopping.

Moreover, Fig. 4 contains the results for MFVRP-F with and without warm-starting from the pretrained solution of MFVRP-G. For the training steps seen in the figure, around 3 minutes of GPU training time were used, see Tables 1 and 2. The pretrained solution provides significant increases in performance for the fine-tuned MFVRP-F solution, in contrast to cold-starting training on a given VRP instance.

As a result of the fast training time using a JAX-based learning implementation, our approach is applicable even in time-critical scenarios despite the very large scale of vehicle routing problems, and therefore may prove advantageous over classical methods such as OR-Tools.

Table 1: Comparison and evaluation of VRP methods on various datasets, each consisting of 50 problems for each configuration $(M, K)$. In the finite dataset, $K = |\mathcal{X}|$, whereas in uniform it is a hyperparameter of our methods, and in cities it is also the number of selected city locations. The confidence intervals for the objectives are in the range of $< 1$.

| | Method | $M = 100$, $K = 3$ | | | $M = 1000$, $K = 3$ | | | $M = 5000$, $K = 3$ | | | $M = 50000$, $K = 3$ | | |
| | | Obj. | Gap | Time | Obj. | Gap | Time | Obj. | Gap | Time | Obj. | Gap | Time |
|---|---|---|---|---|---|---|---|---|---|---|---|---|---|
| Finite | Greedy | 154.35 | 55.64 % | 0.06 s | 1548.87 | 61.20 % | 1.02 s | 7715.41 | 58.73 % | 5.52 s | 73700.7 | 52.16 % | 276.79 s |
| | OR-Tools | 100.76 | 1.60 % | 49.98 s | 1056.95 | 9.99 % | 300.00 s | 5311.167 | 9.27 % | 8001.01 s | − | − | − |
| | PyVRP | 99.17 | * | 180.03 s | 960.86 | * | 180.53 s | 4860.81 | * | 192.88 s | − | − | − |
| | MFVRP-G | 107.78 | 8.68 % | 5.09 s | 1056.68 | 9.97 % | 12.24 s | 5250.19 | 7.99 % | 32.69 s | 49692.9 | 2.59 % | 502.47 s |
| | MFVRP-F | 105.72 | 6.61 % | 173.2 s | 1030.66 | 7.26 % | 176.30 s | 5076.18 | 4.43 % | 199.82 s | 48438.2 | * | 788.39 s |
| Cities | Greedy | 79.43 | 55.99 % | 0.06 s | 795.35 | 49.26 % | 0.78 s | 3958.64 | 49.49 % | 4.62 s | 44404.7 | 49.83 % | 278.56 s |
| | OR-Tools | 50.92 | * | 60.00 s | 532.91 | * | 300.00 s | 2647.75 | * | 8000.01 s | − | − | − |
| | PyVRP | 58.63 | 15.14 % | 180.02 s | 576.70 | 8.22 % | 181.02 s | 2879.56 | 8.75 % | 201.64 s | − | − | − |
| | MFVRP-G | 58.09 | 14.08 % | 6.6 s | 551.25 | 3.44 % | 14.39 s | 2721.67 | 2.79 % | 34.45 s | 30743.2 | 3.73 % | 523.10 s |
| | MFVRP-F | 56.90 | 11.74 % | 173.03 s | 544.26 | 2.13 % | 180.66 s | 2687.12 | 1.49 % | 223.08 s | 29635.9 | * | 775.41 s |
| Uniform | Greedy | 217.76 | 73.09 % | 0.06 s | 2206.73 | 87.12 % | 1.02 s | 11032.74 | 89.65 % | 5.62 s | 110313.7 | 72.30 % | 275.73 s |
| | OR-Tools | 131.35 | 4.40 % | 60.00 s | 1230.68 | 4.35 % | 300.00 s | 5919.15 | 1.74 % | 8000.01 s | − | − | − |
| | PyVRP | 125.81 | * | 180.03 s | 1179.32 | * | 180.63 s | 5817.59 | * | 194.11 s | − | − | − |
| | MFVRP-G | 164.99 | 31.14 % | 4.92 s | 1406.12 | 19.23 % | 9.73 s | 6648.92 | 14.29 % | 31.77 s | 64394.3 | 0.58 % | 494.12 s |
| | MFVRP-F | 164.71 | 30.92 % | 159.38 s | 1406.39 | 19.26 % | 172.63 s | 6633.06 | 14.02 % | 193.26 s | 64024.5 | * | 727.07 s |

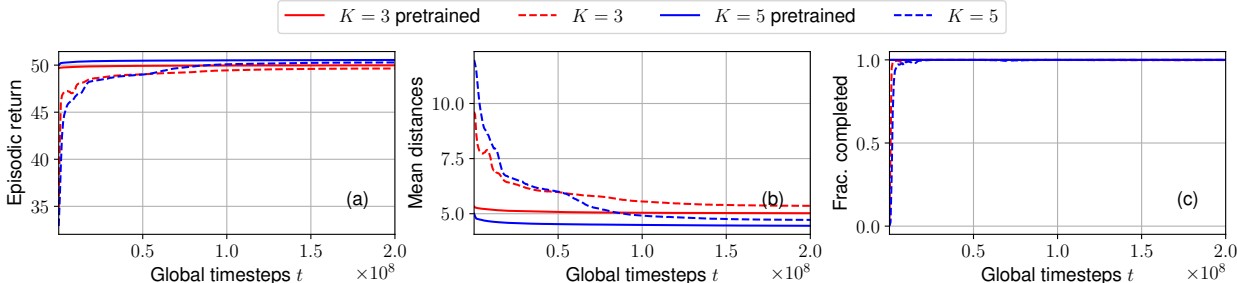

Figure 4: Training curves for MFVRP-F (single trial). Comparison of returns, travelled distances, episode lengths and percentage of completed episodes for MFVRP-F on a particular VRP instance, for $K = 3, M = 500$ and $K = 5, M = 1000$.

**Quantitative comparison of speed.** In Table 1 and Table 2, we quantitatively compare the results of our approach against OR-Tools and PyVRP as strong practical baselines.

We observe that both the general MFVRP-G and our fine-tuned MFVRP-F (including training time) are faster than OR-Tools in large scenarios, which was configured with maximum computation times of 1 to over 120 minutes for small and larger problem configurations respectively. Training was performed on a GPU, but evaluation was performed without. For MFVRP-G the time is the feedforward evaluation time on a standard CPU, whereas for MFVRP-F the times also include the training time on the GPU. Meanwhile, we have slight variation in training and postprocessing time taken, due to minor performance fluctuations. Both PyVRP and OR-Tools failed to produce any solution for $M = 50000$ within our time limit of more than 120 minutes. For the former, this is largely due to the pre-processing bottleneck of constructing the complete $M \times M$ distance matrix, a step with quadratic complexity that becomes intractable at this scale. Therefore, their results are omitted from the tables for this instance size.

Overall, we conclude that in terms of speed and scalability, the MF approach can be advantageous for very large scales. Moreover, we would like to point out that most of the increase in time over increasing $M$ for MFVRP solutions is due to inefficient Python postprocessing code where objects (also leftover objects) are assigned to vehicles. This increase in time can be made significantly more efficient. We point out that MFVRP therefore scales to near-arbitrarily many vehicles if required, with speeds nearly matching simple heuristics.

Table 2: Comparison and evaluation of VRP methods for $K = 5$ as in Table 1.

| | Method | $M = 100, K = 5$ | | | $M = 1000, K = 5$ | | | $M = 5000, K = 5$ | | | $M = 50000, K = 5$ | | |
|---|---|---|---|---|---|---|---|---|---|---|---|---|---|
| | | Obj. | Gap | Time | Obj. | Gap | Time | Obj. | Gap | Time | Obj. | Gap | Time |
| Finite | Greedy | 182.28 | 78.62 % | 0.06 s | 1849.05 | 84.11 % | 1.02 s | 9208.56 | 83.27 % | 5.64 s | 86675.4 | 68.63 % | 258.47 s |
| | OR-Tools | 113.00 | 10.73 % | 57.12 s | 1174.35 | 16.93 % | 300.00 s | 5978.828 | 18.99 % | 8001.01 s | – | – | – |
| | PyVRP | 102.05 | * | 180.03 s | 1004.32 | * | 180.49 s | 5024.67 | * | 191.90 s | – | – | – |
| | MFVRP-G | 130.91 | 28.28 % | 10.43 s | 1222.37 | 21.72 % | 14.99 s | 6029.62 | 20.00 % | 42.79 s | 55017.1 | 7.03 % | 507.06 s |
| | MFVRP-F | 128.68 | 26.10 % | 215.68 s | 1172.54 | 16.75 % | 217.8 s | 5766.29 | 14.76 % | 241.85 s | 51401.0 | * | 859.22 s |
| Cities | Greedy | 105.87 | 68.69 % | 0.08 s | 1064.14 | 74.58 % | 0.76 s | 5333.94 | 75.25 % | 4.32 s | 51858.1 | 61.84 % | 258.02 s |
| | OR-Tools | 64.70 | 3.09 % | 60.00 s | 670.29 | 9.97 % | 300.00 s | 3326.19 | 9.29 % | 8000.01 s | – | – | – |
| | PyVRP | 62.76 | * | 180.02 s | 609.53 | * | 180.66 s | 3043.34 | * | 198.39 s | – | – | – |
| | MFVRP-G | 79.61 | 26.84 % | 13.29 s | 729.45 | 19.67 % | 18.76 s | 3581.63 | 17.69 % | 53.36 s | 34455.7 | 7.53 % | 513.36 s |
| | MFVRP-F | 78.92 | 25.75 % | 202.17 s | 731.32 | 19.98 % | 207.97 s | 3563.57 | 17.09 % | 228.8 s | 32042.1 | * | 798.19 s |
| Uniform | Greedy | 217.76 | 73.09 % | 0.06 s | 2206.73 | 87.13 % | 1.02 s | 11032.74 | 89.65 % | 5.58 s | 110313.7 | 72.28 % | 262.77 s |
| | OR-Tools | 131.35 | 4.40 % | 60.00 s | 1230.67 | 4.36 % | 300.00 s | 5919.67 | 1.74 % | 8000.01 s | – | – | – |
| | PyVRP | 125.81 | * | 180.02 s | 1179.24 | * | 180.73 s | 5818.09 | * | 194.29 s | – | – | – |
| | MFVRP-G | 171.04 | 35.95 % | 9.64 s | 1458.49 | 23.68 % | 14.74 s | 6836.91 | 17.51 % | 43.35 s | 66644.9 | 4.08 % | 488.32 s |
| | MFVRP-F | 171.91 | 36.64 % | 190.95 s | 1453.92 | 23.30 % | 174.96 s | 6759.07 | 16.17 % | 192.39 s | 64029.3 | * | 806.69 s |

**Quantitative comparison of objectives.** As for the quality of objectives given by the total distance travelled by all vehicles, we find that MFVRP-F is usually better than MFVRP-G as a result of fine-tuning on the particular VRP instance of interest. In a separate test, we verify that MFVRP-FD gives results in-between MFVRP-F and MFVRP-G (106.72 for $M = 100, K = 3$).

In comparison to baselines, we find that MFVRP usually outperforms simple heuristics but is behind OR-Tools and PyVRP in smaller-sized problem instances. A key question is why MFVRP, even with fine-tuning, is outperformed by solvers like PyVRP on smaller instances. The reason lies in the nature of our approximation. MFVRP optimizes a discretized, mean-field version of the problem, which is an accurate proxy only when the number of agents $N$ is very large. For smaller $N$, 1. the stochasticity of the finite system causes the empirical distributions to deviate significantly from the smooth mean-field dynamics, an error that persists throughout the episode. Furthermore, 2. the clustering introduces a discretization error that is more impactful on shorter total routes found in smaller problems. In contrast, solvers like PyVRP use heuristics like HGS that perform sophisticated local search directly on the exact, discrete problem graph, which is highly effective at these scales. While fine-tuning MFVRP helps, it cannot fully bridge this fundamental gap between optimizing an approximate model versus searching on the exact problem. We also empirically show the significance of the MF error for small-to-medium instances in Appendix C.1.

Moreover, we attribute significant suboptimality to the limited efficiency of the RL solution. In particular, we note that the RL solutions shown in Figure 3 are unstable, i.e. the solution begins to deteriorate at a certain point and does not stay in a local optimum. Further work invested into reward shaping, hyperparameter tuning and neural network architecture may improve results further. This is supported by the limited RL performance for higher $K$ as seen in Appendix C.1.

As the size of problems increases however, the gap to the best method grows increasingly small. We also find that MFVRP can outperform OR-Tools and PyVRP under the right circumstances ($M = 5000$, "finite" and "cities" respectively). Moreover, the method scales automatically to arbitrary numbers of vehicles and objects, whereas OR-Tools and PyVRP take very long to produce results at very large scales. Moreover, we find that for $M = 1000$ and $M = 5000$ in the finite dataset, our approach achieves an increase in performance over OR-Tools. This shows that a mean-field-based methodology to optimization problems can in fact outperform more standard methods in the case of (a) accurate clustering ("finite" dataset), and (b) high numbers of vehicles.

**Comparison between datasets.** Comparing between different scenarios and parameters, we find an increasingly good approximation of the MFVRP solution as the number of vehicles and objects becomes large, $M \to \infty$. In general, the gap between MFVRP and baseline performance becomes increasingly small or disappears as $M$ increases.

Additionally, we can see that the suboptimality of MFVRP is strongly associated with the clusteredness of the dataset as expected. In the "finite" dataset, we find significantly better comparative performance due to the finite space of locations $|\mathcal{X}| = K$, whereas in "cities" the sources and destinations are scattered around the city centroids. Finally, in "uniform" we essentially have the worst-case for clustering-based approximations, with accordingly worse gaps in performance. To further improve the gap stemming from discretization however, we have to increase $K$, which in turn increases the hardness of the associated RL problem, since the action space is of dimension $K^2$. Scaling up RL towards very high-dimensional continuous action spaces is difficult and left as future work, as our initial experiments with low-rank approximations of $T_t$ and higher $K > 5$ in Appendix C.1 show (neither experiment is successful in achieving satisfactory performance).

## 5   Discussion and Conclusion

In this work, we have proposed a scalable RL methodology based on MFC for solving large-scale CVRP with pickup and delivery. The proposed methods were not only theoretically motivated, but also empirically evaluated and compared against existing baselines as a proof of concept. While the general case with many clusters is yet difficult due to the unique challenge of high-dimensional actions, the solution of particular cases remains useful. In particular, existing standard methods such as OR-Tools may take too long to find a solution for many vehicles and locations (see Table 2), while our method provides a fast solution. While

solving high-dimensional continuous RL such as the MFVRP remains difficult, we still obtain RL complexities independent of the number of clients (or linear by sampling a VRP solution and routes for each agent from the limiting solution, which is the minimal asymptotic complexity). Overall, we have investigated the applicability of mean-field methods to discrete optimization settings and found that it is at the very least feasible.

At the same time, we have also seen the limitations in applying mean-field methods in standard discrete optimization settings (e.g., Section 2.5 and experiments). In future work, one could consider how to apply MFC-based solutions for general non-pickup-and-delivery VRP and other hard combinatorial optimization problems. One could also scale up to more clusters, which is challenging due to the $K^2$-dimensional action space. Promising directions for this include curriculum learning, energy-based actor-critic methods (Haarnoja et al., 2018), or using structured representations of the transport map $T_t$, such as low-rank factorizations. Further, it remains to be studied whether one can apply appropriate inductive biases via graph neural networks to generalize to any number of clusters. Finally, one may consider taking inspiration from related classical complexity reduction such as the fast multipole method (Rokhlin, 1985), or avoiding discretization altogether by suitable kernel-based parametrizations.

## Impact Statement

This paper presents work whose goal is to advance the field of Machine Learning. There are many potential societal consequences of our work, none which we feel must be specifically highlighted here.

## Acknowledgements

This research has been partially funded by the German Federal Ministry of Education and Research (BMBF) through grant 01IS23067 Software Campus 3.0 Technische Universität Darmstadt as part of the Software Campus project 'RL4MFRP', and by the LOEWE initiative (Hesse, Germany) within the emergenCITY center. The authors acknowledge Lichtenberg high performance computer of the TU Darmstadt for providing computational facilities for the calculations for this research.

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

# A  Dataset Details

In this study, we generate 3 different types of datasets, called uniform, finite and cities. For each type of dataset we vary the amount of objects to be delivered $M = \{100, 1000, 5000, 50000\}$, and for the finite and cities distribution we also vary the amount of clusters, $K \in \{3, 5\}$.

The positions of the $K$ clusters in finite are sampled uniformly from $[-1, 1]^2$. Meanwhile, in cities we sample the positions from a list of rescaled longitudes and latitudes of the 80 most populated German cities. The rescaling is performed by subtracting the mean longitude and latitude of $(51, 10)$ and dividing by 5. The source and destination cluster assignments are sampled from a flat Dirichlet distribution. The exact position of the $M$ targets and sources is then sampled from a Gaussian distribution around the positions of one of the corresponding $K$ clusters, with a standard deviation of 0 and 0.02 for finite and cities respectively, with samples clipped between -1 and 1. For the latter, this standard deviation of 0.02 corresponds to a city radius (95% of samples inside) of $16\,\mathrm{km}$.

For the uniform dataset we sample $M$ sources and destinations uniformly from $[-1, 1]^2$. We generate 50 instances for each combination of $M$ and $k$, resulting in 200 VRP instances for uniform, and 400 VRP instances for cities and finite.

# B  Proofs

## B.1  Mean-Field Convergence

*Proof of Theorem 2.1.* First, note that with deterministic $\hat{\pi}$ comes constant (non-random) $T_t$ output over all $\mu_t, \nu_t$. Indeed, even constant MFC policies are sufficient due to the deterministic evolution of $\mu_t, \nu_t$ given the initial state, and are equivalent to any trained RL policy $\hat{\pi}$ for the limiting MFC problem with open-loop control sequence $T_0, T_1, T_2, \ldots$. We therefore use an equivalent optimal, but constant MFC policy $\hat{\pi}'_t(T|\cdot, \cdot) = \delta_{T_t}$. Therefore, $T_t$ is no longer random and entirely deterministic.

The statement is shown inductively by convergence in expectation, $\mathbb{E}\left[\|\mu_t^N - \mu_t\|_\infty\right] = \mathcal{O}(1/\sqrt{N}) \to 0$ and $\mathbb{E}\left[\|\nu_t^N - \nu_t\|_\infty\right] = \mathcal{O}(1/\sqrt{N}) \to 0$. Convergence in probability then follows. At time $t = 0$, the statement is immediate by weak law of large numbers (LLN). Assuming the statement at time $t$, then at time $t + 1$, as a short sketch, we obtain

$$
\mathbb{E}\left[\|\mu_{t+1}^N - \mu_{t+1}\|_\infty\right] \leq \sum_{x \in \mathcal{X}} \mathbb{E}\left[\left\|\frac{1}{N}\sum_{\substack{s \in \mathcal{X} \\ i \leq N}} \delta_{x_t^i}(s)u_{t,s}^i - \frac{1}{N}\sum_{\substack{s \in \mathcal{X} \\ i \leq N}} \delta_{x_t^i}(s)T_t(s, x)\right\|\right]
$$
$$
+ \mathbb{E}\left[\left\|\frac{1}{N}\sum_{\substack{s \in \mathcal{X} \\ i \leq N}} \delta_{x_t^i}(s)T_t(s, x) - \sum_{s \in \mathcal{X}} \mu_t(s)T_t(s, x)\right\|\right],
$$

and can apply induction assumption for the latter term, and a weak LLN argument for the former. The same holds for $\nu_t^N, \nu_t$.

More precisely, for the analysis, we lift the actions to a new action space $\mathcal{X}^{\mathcal{X}}$ (Carmona et al., 2018) and note the equivalence *in distribution* to the original MARL system by using $u_{t,x}^i$ whenever vehicle $i$ is in state $x$. Then, we may apply the discussed constant MFC policies by sampling $u_{t,x}^i \sim T_t(x, \cdot)$ for all $x \in \mathcal{X}$ and using the corresponding component, instead of $u_t^i \sim T_t(x_t^i, \cdot)$. The advantage in analysis is that $u_{t,x}^i$ is i.i.d. for all $i, t, x$. Then, $\nu_{t+1}^N(s, d)$ is given by subtracting from $\nu_t^N(s, d)$ the random variables $\frac{1}{N}\mathbf{1}_{x_t^i = s, u_{t,s}^i = d}$ for each of $i \in \mathcal{N}$ vehicles, up to at most $NR\nu_{t+1}^N(s, d)$ many.

The overall result is then obtained in two steps, which are shown in App. B.2 and B.3. The proof proceeds in the above system where the actions are i.i.d., and the results follow for the original system by equivalence *in distribution* of both systems for the two following lemmas.

**Lemma B.1** (Propagation of Chaos I). *Consider finite $\mathcal{X}$, $t_{\max} < \infty$ and deterministic $\hat{\pi}$. For each $t \leq t_{\max}$, as $N \to \infty$,*

$$\mathbb{E}\left[\|\mu_t^N - \mu_t\|_\infty\right] = \mathcal{O}(1/\sqrt{N}) \to 0\,.$$

**Lemma B.2** (Propagation of Chaos II). *Consider finite $\mathcal{X}$, $t_{\max} < \infty$ and deterministic $\hat{\pi}$. For each $t \leq t_{\max}$, as $N \to \infty$,*

$$\mathbb{E}\left[\|\nu_t^N - \nu_t\|_\infty\right] = \mathcal{O}(1/\sqrt{N}) \to 0$$

From the above lemmas, the proof of Theorem 2.1 follows: The desired statement was shown inductively by convergence in expectation, $\mathbb{E}\left[\|\mu_t^N - \mu_t\|_\infty\right] = \mathcal{O}(1/\sqrt{N}) \to 0$ and $\mathbb{E}\left[\|\nu_t^N - \nu_t\|_\infty\right] = \mathcal{O}(1/\sqrt{N}) \to 0$ by Lemmas B.1 and B.2. Convergence in probability then follows by Van der Vaart (2000, Theorem 2.7) and deterministic $\mu_t, \nu_t$. $\qquad\square$

## B.2 Propagation of Chaos I

*Proof of Lemma B.1.* For the analysis, we lift the actions to a new action space $\mathcal{X}^{\mathcal{X}}$ (Carmona et al., 2018) and note the equivalence *in distribution* to the original MARL system, as discussed in B.1. We prove the statement via induction over $t$. For $t = 0$, the statement follows from a law of large numbers argument (see induction step below, first term). Thus, it remains to show the induction step using the induction assumption (IA). Assume that the equation holds at time $t$. Then, at time $t+1$ we have

$$\mathbb{E}\left[\|\mu_{t+1}^N - \mu_{t+1}\|_\infty\right] = \mathbb{E}\left[\max_{x \in \mathcal{X}} |\mu_{t+1}^N(x) - \mu_{t+1}(x)|\right]$$

$$\leq \mathbb{E}\left[\sum_{x \in \mathcal{X}} |\mu_{t+1}^N(x) - \mu_{t+1}(x)|\right] = \sum_{x \in \mathcal{X}} \mathbb{E}\left[|\mu_{t+1}^N(x) - \mu_{t+1}(x)|\right]$$

$$= \sum_{x \in \mathcal{X}} \mathbb{E}\left[\left|\left(\frac{1}{N}\sum_{s \in \mathcal{X}}\sum_{i \leq N} \delta_{x_t^i}(s) u_{t,s}^i(x)\right) - \sum_{s \in \mathcal{X}} \mu_t(s) T_t(s,x)\right|\right]$$

$$= \sum_{x \in \mathcal{X}} \mathbb{E}\left[\left|\left(\frac{1}{N}\sum_{s \in \mathcal{X}}\sum_{i \leq N} \delta_{x_t^i}(s) u_{t,s}^i(x)\right) - \left(\frac{1}{N}\sum_{s \in \mathcal{X}}\sum_{i \leq N} \delta_{x_t^i}(s) T_t(s,x)\right)\right.\right.$$

$$\left.\left. + \left(\frac{1}{N}\sum_{s \in \mathcal{X}}\sum_{i \leq N} \delta_{x_t^i}(s) T_t(s,x)\right) - \sum_{s \in \mathcal{X}} \mu_t(s) T_t(s,x)\right|\right]$$

$$\leq \sum_{x \in \mathcal{X}} \mathbb{E}\left[\left|\left(\frac{1}{N}\sum_{s \in \mathcal{X}}\sum_{i \leq N} \delta_{x_t^i}(s) u_{t,s}^i(x)\right) - \frac{1}{N}\sum_{s \in \mathcal{X}}\sum_{i \leq N} \delta_{x_t^i}(s) T_t(s,x)\right|\right]$$

$$+ \mathbb{E}\left[\left|\left(\frac{1}{N}\sum_{s \in \mathcal{X}}\sum_{i \leq N} \delta_{x_t^i}(s) T_t(s,x)\right) - \sum_{s \in \mathcal{X}} \mu_t(s) T_t(s,x)\right|\right]$$

$$= \sum_{x \in \mathcal{X}} \mathbb{E}\left[\left|\left(\frac{1}{N}\sum_{s \in \mathcal{X}}\sum_{i \leq N} \delta_{x_t^i}(s) u_{t,s}^i(x)\right) - \frac{1}{N}\sum_{i \leq N} \mathbb{E}\left[u_t^i(x)\Big| x_t^1, \ldots, x_t^N\right]\right|\right]$$

$$+ \mathbb{E}\left[\left|\sum_{s \in \mathcal{X}} T_t(s,x)\left(\mu_t(s) - \frac{1}{N}\sum_{i \leq N} \delta_{x_t^i}(s)\right)\right|\right]$$

$$\leq \sum_{x \in \mathcal{X}} \mathbb{E}\left[\frac{1}{N^2}\left(\sum_{i \leq N} u_t^i(x) - \mathbb{E}\left[u_t^i(x)\Big| x_t^1, \ldots, x_t^N\right]\right)^2\right]^{1/2} + \sum_{s \in \mathcal{X}} T_t(s,x) \underbrace{\mathbb{E}\left[\left|\mu_t(s) - \frac{1}{N}\sum_{i \leq N} \delta_{x_t^i}(s)\right|\right]}_{\leq \mathcal{O}(1/\sqrt{N}) \text{ by IA}}$$

$$= \sum_{x \in \mathcal{X}} \mathbb{E}\left[\frac{1}{N^2} \sum_{i \leq N} \left(u_t^i(x) - \mathbb{E}\left[u_t^i(x) \Big| x_t^1, \ldots, x_t^N\right]\right)^2\right]^{1/2} + \mathcal{O}(1/\sqrt{N})$$

$$= \frac{2|\mathcal{X}|}{\sqrt{N}} + \mathcal{O}(1/\sqrt{N}),$$

which concludes the proof. □

### B.3 Propagation of Chaos II

*Proof of Lemma B.2.* For the analysis, we lift the actions to a new action space $\mathcal{X}^{\mathcal{X}}$ (Carmona et al., 2018) and note the equivalence *in distribution* to the original MARL system, as discussed in B.1. As before, we prove the Lemma via induction over $t$ and note that the induction start $t = 0$ holds by a law of large numbers argument (alternatively, see induction step below). Now, the induction step is provided by the following argument, where we assume that the statement holds for $t$:

$$\mathbb{E}\left[\|\nu_{t+1}^N - \nu_{t+1}\|_\infty\right] = \mathbb{E}\left[\max_{s,d \in \mathcal{X}} |\nu_{t+1}^N(s,d) - \nu_{t+1}(s,d)|\right]$$

$$\leq \mathbb{E}\left[\sum_{s,d \in \mathcal{X}} |\nu_{t+1}^N(s,d) - \nu_{t+1}(s,d)|\right] = \sum_{s,d \in \mathcal{X}} \mathbb{E}\left[|\nu_{t+1}^N(s,d) - \nu_{t+1}(s,d)|\right]$$

$$= \sum_{s,d \in \mathcal{X}} \mathbb{E}\left[\left|\max\left(0, \underbrace{\nu_t^N(s,d) - \frac{1}{RN}\sum_{i \leq N} \delta_{x_t^i}(s) u_{t,s}^i(d)}_{=:\mathrm{I}_N}\right) - \max(0, \underbrace{\nu_t(s,d) - \mu_t(s)T_t(s,d)/R}_{=:\mathrm{II}})\right|\right] \quad (5)$$

For the next argument, fix some arbitrary tuple $(s,d) \in \mathcal{X}^2$. For the moment, focus on

$$\mathbb{E}\left[\left|\nu_t^N(s,d) - \frac{1}{RN}\sum_{i \leq N} \delta_{x_t^i}(s) u_{t,s}^i(d) - (\nu_t(s,d) - \mu_t(s)T_t(s,d)/R)\right|\right]$$

$$\leq \mathbb{E}\left[|\nu_t^N(s,d) - \nu_t(s,d)|\right] + \frac{1}{R}\mathbb{E}\left[\left|\mu_t(s)T_t(s,d) - \frac{1}{N}\sum_{i \leq N} \delta_{x_t^i}(s) u_{t,s}^i(d)\right|\right] = \mathcal{O}(1/\sqrt{N}),$$

where the first summand converges to zero by the induction assumption and the second summand converges to zero by an argument as in the proof of Theorem B.1. The above result implies convergence of equation (5) to zero because for each arbitrary but fixed tuple $(s,d) \in \mathcal{X}^2$ and any $N \in \mathbb{N}$ we have

$$|\max(0, \mathrm{I}_N) - \max(0, \mathrm{II})| \leq |\mathrm{I}_N - \mathrm{II}|.$$

To see that the inequality is true, make a case distinction. If both $\mathrm{I}_N$ and $\mathrm{II}$ are non-negative, the statement trivially holds. If both $\mathrm{I}_N$ and $\mathrm{II}$ are negative, the left side of the inequality is 0 and thereby the inequality is obviously fulfilled since there is an absolute value on the right side. Finally, in the third case, assume that $\mathrm{I}_N$ is non-negative and $\mathrm{II}$ is negative (or vice versa). In the third case, the left side equals $\mathrm{I}_N$ while the right side equals $\mathrm{I}_N - \mathrm{II} > \mathrm{I}_N$ since $\mathrm{II}$ is negative. This completes the proof of Theorem B.2. □

### B.4 Approximate Optimality with High Probability

*Proof of Corollary 2.2.* The proof idea is to compare the objectives (sums of expected rewards) of the respective policies and systems, and thereby establish the following sequence of inequalities for objectives, for any $\delta > 0$ holding at least with probability $1 - \delta$,

$$\mathrm{MARL} \overset{(1)}{\geq} \mathrm{MFC}^* - \Delta_1 \overset{(2)}{\geq} \mathrm{MFC}'^* - \Delta_1 - \Delta_2 \overset{(3)}{\geq} \mathrm{MARL}^* - \Delta_1 - \Delta_2$$

with $\Delta_1 = \mathcal{O}\left(\frac{1}{\delta\sqrt{N}}\right)$ and $\Delta_2 = \mathcal{O}\left(\frac{1}{\delta\sqrt{N}}\right)$, where MARL denotes the objective of the sampled MARL solution, MARL$^*$ is the optimal MARL objective, MFC$^*$ is the MFC solution's objective from which the MARL solution is sampled, and MFC$'^*$ is the optimal MFC objective with starting conditions matching the empirical sampled VRP instance $\nu_0^N, \mu_0^N$.

**Term 1.** Consider an optimal MFC solution where the corresponding objective value is denoted by MFC$^*$ in the above inequality sequence. By Theorem 2.1, the probability of any positive fraction of objects $\%_{\text{rem}} = \sum_{s,d} \nu_{t_{\max}}^N(s,d) \geq 0$ remaining (more than in the MFVRP) after termination time $t_{\max}$ of the MFVRP MDP being larger than some $\varepsilon > 0$ is

$$\mathbb{P}\left(\%_{\text{rem}} \geq \varepsilon\right) = \mathbb{P}\left(\left|\sum_{s,d} \nu_{t_{\max}}^N(s,d) - \sum_{s,d} \nu_{t_{\max}}(s,d)\right| \geq \varepsilon\right)$$

$$\leq \mathbb{E}\left[\left|\sum_{s,d} \nu_{t_{\max}}^N(s,d) - \sum_{s,d} \nu_{t_{\max}}(s,d)\right|\right] \cdot \frac{1}{\varepsilon}$$

by Markov inequality. From Theorem 2.1, $\mathbb{E}[\|\mu_t^N - \mu_t\|_\infty] \leq C_t/\sqrt{N}$ and $\mathbb{E}[\|\nu_t^N - \nu_t\|_\infty] \leq D_t/\sqrt{N}$. Let $C_{\max} = \max_t(C_t + D_t)$. Then the expected difference is bounded:

$$\mathbb{E}\left[\left|\sum_{s,d} \nu_{t_{\max}}^N(s,d) - \sum_{s,d} \nu_{t_{\max}}(s,d)\right|\right] \leq L_r \cdot \frac{C_{\max}}{\sqrt{N}} \cdot |\mathcal{X}|^2.$$

Letting the probability of remaining objects be at most $\frac{\delta}{4}$ and solving for $\epsilon$, we have

$$\varepsilon = \frac{4 L_r C_{\max} |\mathcal{X}|^2}{\delta\sqrt{N}}$$

as an optimal MFC solution transports all objects in the limit, $\sum_{s,d} \nu_{t_{\max}}(s,d) = 0$.

Similarly, by Theorem 2.1, the MARL trajectories' rewards are close to the MFC rewards up to time $t_{\max}$ with high probability,

$$\mathbb{P}\left(\left|\sum_{t \leq t_{\max}} r_t^N - \sum_{t \leq t_{\max}} r_t\right| \geq \varepsilon\right) \leq \mathbb{E}\left[\left|\sum_{t \leq t_{\max}} r_t^N - \sum_{t \leq t_{\max}} r_t\right|\right] \cdot \frac{1}{\varepsilon} \leq \frac{t_{\max} C_{\max}}{\varepsilon\sqrt{N}}.$$

Again, letting the right-hand side be at most $\frac{\delta}{4}$, we get

$$\varepsilon = \frac{4 t_{\max} C_{\max}}{\delta\sqrt{N}}.$$

Let the former event be $A = \{\%_{\text{rem}} \geq \varepsilon\}$, and the latter $B = \{\left|\sum_{t \leq t_{\max}} r_t^N - \sum_{t \leq t_{\max}} r_t\right| \geq \varepsilon\}$. Overall, therefore $\mathbb{P}(\bar{A} \wedge \bar{B}) \geq 1 - \frac{\delta}{2}$. Moreover, $\bar{A} \wedge \bar{B}$ jointly implies that a sampled solution's return is approximately given by the optimal MFC solution's return up to a difference

$$\Delta_1 = \frac{4 L_r C_{\max} |\mathcal{X}|^2}{\delta\sqrt{N}} \cdot 2\operatorname{diam}(\mathcal{X}) + \frac{4 t_{\max} C_{\max}}{\delta\sqrt{N}}$$

since each remaining object takes at most $2\operatorname{diam}(\mathcal{X})$ to transport, and thus establishes the first inequality (1).

**Term 3.** To prove the third inequality (3), note that the return of any optimal MARL solution can also be exactly obtained in a MFC system with modified initial distributions, by using $\nu_0^N, \mu_0^N$ as starting values

instead of $\nu_0, \mu_0$. Letting $T_t(s, d) = \frac{\sum_i \mathbf{1}_s(x_t^i) \mathbf{1}_d(u_t^i)}{\sum_i \mathbf{1}_s(x_t^i)}$ obtains exactly the optimal MARL objective in the modified MFC system denoted as MFC$'^*$ in the above inequality sequence. Therefore, the optimal MARL objective is bounded by the optimal modified MFC objective which corresponds to inequality (3).

**Term 2.** Finally, it remains to establish inequality (2). We point out that the optimal MFC objective is Lipschitz continuous in its starting conditions, since Eqs. (2)–(4) are Lipschitz continuous. Let $L_J > 0$ be the corresponding Lipschitz constant. Then

$$\mathbb{P}\left(L_J(\|\mu_0^N - \mu_0\|_\infty + \|\nu_0^N - \nu_0\|_\infty) \geq \varepsilon\right)$$

$$\leq \mathbb{E}\left[L_J(\|\mu_0^N - \mu_0\|_\infty + \|\nu_0^N - \nu_0\|_\infty)\right]\frac{1}{\varepsilon} \leq L_J \frac{4|\mathcal{X}|}{\sqrt{N}} \cdot \frac{1}{\varepsilon}$$

and letting the probability not exceed probability $\frac{\delta}{2}$, we get $\varepsilon = \frac{4L_J|\mathcal{X}|}{\delta\sqrt{N}}$. This implies that we have a bound between optimal modified MFC objective (with changed starting conditions) and the optimal MFC objective

$$\Delta_2 = \frac{4L_J|\mathcal{X}|}{\delta\sqrt{N}},$$

which yields inequality (2).

Therefore, putting the results together, the probability that the sampled MARL solution is $(\Delta_1 + \Delta_2)$-optimal in the MARL system also is at least $1 - \delta$. $\qquad\square$

### B.5 Approximate Optimality under Fine Discretization

*Proof of Theorem 3.1.* The proof structure consists of establishing the following five inequalities of objectives as in Corollary 2.2,

$$\text{MARL} \geq \delta\text{-MARL} - \varepsilon'$$
$$\geq \text{MFC}^* - \varepsilon' - \Delta_1 \geq \text{MFC}'^* - \varepsilon' - \Delta_1 - \Delta_2$$
$$\geq \delta\text{-MARL}^* - \varepsilon' - \Delta_1 - \Delta_2 \geq \text{MARL}^* - 2\varepsilon' - \Delta_1 - \Delta_2.$$

First, we obtain the center inequality sequence

$$\delta\text{-MARL} - \varepsilon' \geq \text{MFC}^* - \varepsilon' - \Delta_1 \geq \text{MFC}'^* - \varepsilon' - \Delta_1 - \Delta_2 \geq \delta\text{-MARL}^* - \varepsilon' - \Delta_1 - \Delta_2$$

by the same argument as in the proof of Corollary 2.2, where $\delta$-MARL refers to the discretized VRP problem.

Therefore, it remains to establish the first and last inequality of the initial inequality sequence. These two inequalities intuitively build on the observation that the discretized $\delta$-MARL system is close to the initial, continuous MARL system. Formally, we know that each of the $M$ objects 'cause' at most two vehicle moves: one to be picked up and one to be delivered to the destination. In the worst case, all $N$ vehicles have to additionally fly back to their respective depots, which upper bounds the total number of travels by $2M + N$.

By the triangle inequality, we then know that each travel in the discretized system deviates at most $2\max_k \text{diam}(C_k)$ from the respective travel in the continuous system. Consequently, the overall sum of deviations, and therefore the deviation of objective in the systems, is upper bounded by $\varepsilon' = 2 \cdot (2M + N)\max_k \text{diam}(C_k)$. This argument holds uniformly for all solutions and hence establishes

$$\text{MARL} \geq \delta\text{-MARL} - \varepsilon' \text{ and } \delta\text{-MARL}^* \geq \text{MARL}^* - \varepsilon'$$

and thereby concludes the proof. $\qquad\square$

## C   Additional Experiments

In this section, we perform additional experiments on low-rank approximations to reduce the complexity of the RL problem, as well as run for higher $K$.

### C.1 Low-Rank Approximation

In Fig. 5, we can see the results for a low-rank approximation of $T_t$, where we essentially reduce the dimensionality of action space from $K^2$ to $K$ using a rank-1 approximation. Essentially, each column of $T_t$ is the same, because $T_t$ is row stochastic and therefore rows are normalized. As can be seen from the figure, the rank-1 approximation is insufficiently expressive and greatly reduces the achieved objectives and distances travelled in comparison to the full-rank version.

### C.2 Tests on higher K

In Fig. 5, we can see the results for higher $K$. It can be seen that the RL algorithms do not converge to good solutions. We used the exact same settings and implementation as for $K = 5$ and $K = 3$. This demonstrates the scalability challenge posed by the $K^2$-dimensional action space. High-dimensional continuous-action RL still remains very challenging, and this constitutes a limitation of our work.

### C.3 Fractions of leftover objects

As can be seen in Fig. 7, the number of leftover objects at the end of executing the limiting MF actions in the corresponding finite VRPs are significant for smaller $N$, but quickly tend to zero for sufficiently large $N$. This implies that the heuristic for handling any leftover objects may play an important role for suboptimal performance of MFVRP in smaller instances, but its choice is not important for sufficiently large VRP instances.

### C.4 Expected error between empirical and limiting MF

In Fig. 8 we can see that the expected $L_1$ error between the limiting MF and the empirical MF starts at zero at decision epoch zero, since all agents start at the depot and the MF of objects is initialized from the finite problem. After executing a few sampled actions in decision epochs however, we find that the MFs of the limiting system and the sampled MFVRP system will diverge, less so the larger the VRP instance is. It can also be seen that the deviations still remain of significant order (more than a few percent) unless the number of agents is significant (e.g., $\geq 1000$). We can hence see that MFVRP obtains potentially significant suboptimality from its approximation error, unless the VRP instances are of sufficient size.

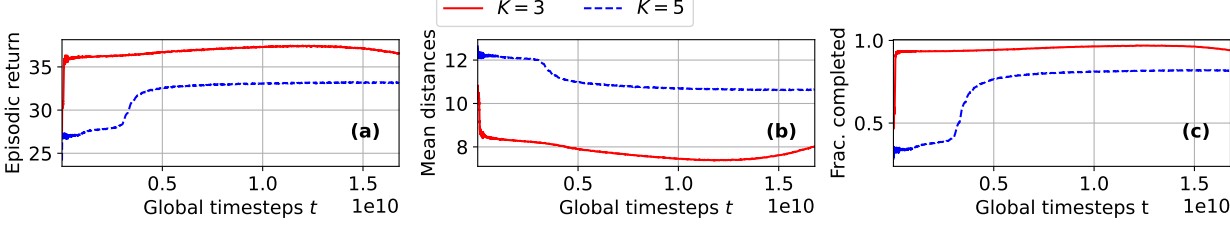

Figure 5: Training curves for MFVRP-G as in Fig. 3, but using a rank-1 approximation for $T_t$, i.e. reducing the dimensionality of action space from $K^2$ to $K$.

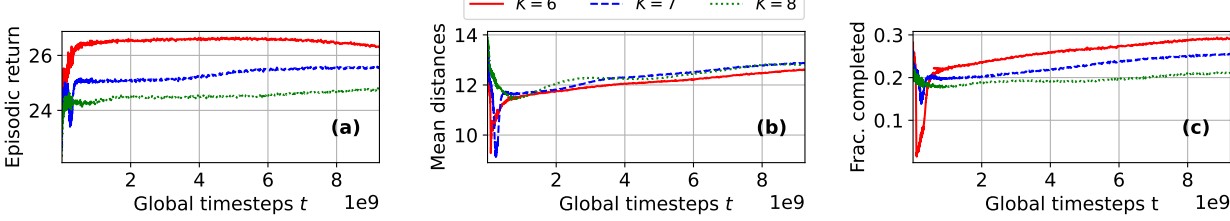

Figure 6: Training curves for MFVRP-G as in Fig. 3, but for higher $K$.

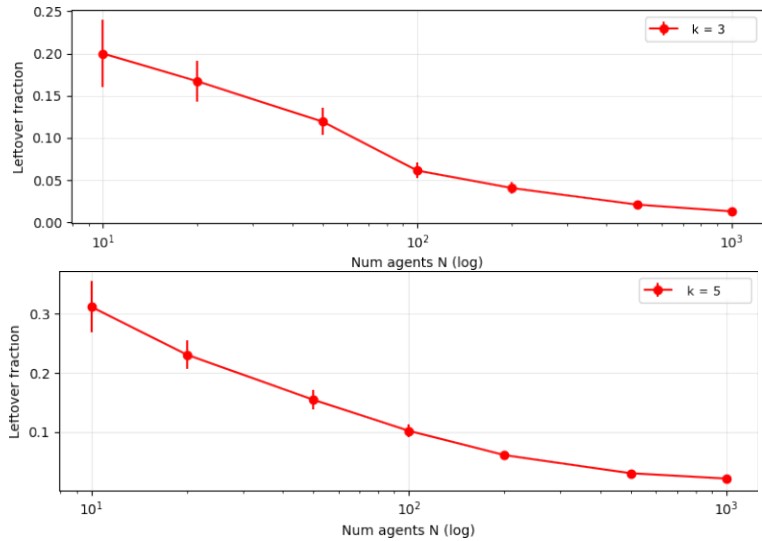

Figure 7: Expected fraction of leftover objects over the size of the sampled VRP problem, together with 95% confidence interval over a total of 50 trials.

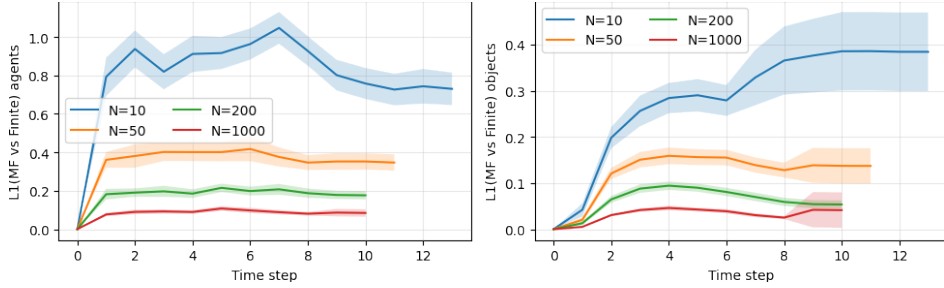

Figure 8: Expected $L_1$ error between limiting and empirical MF, with 95% confidence over 50 trials. (Left: MF of agents; Right: MF of objects; last value extended to longest trial if lengths mismatch)

