# OpenReview forum: "Mean-Field RL for Large-Scale Unit-Capacity Pickup-and-Delivery Problems"
_TMLR — Accepted by TMLR_

### Review · Reviewer_NtR6 · 2025-07-27

**Summary Of Contributions:**

This work proposes to combine mean field control and reinforcement learning to handle large-scale vehicle routing problems.

### Strengths:
- The introduction of mean field control for vehicle routing problems (VRPs) is a novel and interesting approach, particularly for large-scale scenarios.

### Weaknesses:
 - Performance Concerns: The proposed algorithm underperforms baseline methods (e.g., OR-Tools, PyVRP) on small and mid-sized VRPs. While it scales to 50,000-node problems, the solution quality remains unclear—outperforming a naive greedy heuristic is insufficient to demonstrate superiority. Alternative baselines like Hexaly should be considered for a more rigorous comparison.

 - Overly Simplified Assumptions: The assumption of single-capacity vehicles ("carrying capacity of one object") significantly weakens the model’s applicability, as it ignores a key feature of VRPs: multi-pickup/delivery scenarios.

  - Questionable Practical Relevance: Mean field control relies on a large number of agents (vehicles), yet real-world VRPs typically involve limited fleets. The authors should justify this choice with a motivating example or application where such scalability is realistic and necessary.

**Audience:**

Yes

**Audience Explanation:**

Novel methods for VRPs are worth studying.

**Claims And Evidence:**

No

**Claims Explanation:**

See the weakness part.

**Requested Changes:**

1. **Baseline Specification**
   - The authors use "OR-Tools" as a baseline without specifying which specific algorithm(s) were employed. Given OR-Tools' diverse algorithmic offerings, this omission affects reproducibility.

2. **Benchmark Enhancement**
   - The evaluation would benefit from including standard **VRP with Pickup and Delivery (VRPPD)** benchmarks, particularly:
     - https://neo.lcc.uma.es/vrp/vrp-instances/vehicle-routing-problem-with-pick-up-and-deliveries/
   - These would better validate the method's handling of precedence constraints and mixed load scenarios.

3. **Comparative Analysis**
   The baseline comparison should be expanded to include:
   - **Commercial Solvers**:
     - [Hexaly](https://www.hexaly.com/) (notable for its optimization speed)
     - NVIDIA's [cuOpt](https://docs.nvidia.com/cuopt/) (for GPU-accelerated routing)
   - **Learning-Based Methods**: All relevant RL/learning approaches cited in the paper

---

> ### Author Response · Authors · 2025-08-17
>
> We thank Reviewer NtR6 for their valuable feedback and for acknowledging the novelty of our approach. We address the raised weaknesses and requested changes below.
>
> **Concerns/Weaknesses:**
>
> 1.  **Performance Concerns (small/mid-sized VRPs and baselines):** We will clarify that our 'greedy' baseline is a first-order local search (PATH_CHEAPEST_ARC in OR-Tools), a standard and computationally feasible heuristic for the N=50,000 scale. Solvers like Hexaly are further proprietary commercial tools. Instead, our goal is to introduce a new algorithmic paradigm for extreme-scale problems, making open-source scalable heuristics and fast methods like PyVRP the most relevant academic comparison.
> 2.  **Oversimplified Assumptions (unit-capacity):** Unit-capacity was chosen to establish the foundational framework. Extending to higher capacities is conceptually possible as discussed, but increases state space complexity exponentially. This is a limitation and an important future research direction.
> 3.  **Questionable Practical Relevance (large number of agents):** Our work is motivated by emerging large-scale logistics problems with thousands of agents, such as ride-sharing, food delivery, and last-mile gig-economy delivery services (e.g., Amazon Flex). We added a discussion of these motivating applications to the Introduction.
>
> **Requested Changes:**
>
> 1.  **Baseline Specification (OR-Tools):** We used the default OR-Tools routing solver, which employs advanced heuristics and metaheuristics to automatically pick an algorithm. We added this detail to the implementation section.
> 2.  **Benchmark Enhancement (VRPPD datasets):** We agree on the value of standard VRPPD benchmarks. However, our current focus is on unprecedented problem sizes, while the referenced dataset consists of only $M=100$ locations. Moreover, we did not focus on precedence constraints and mixed load scenarios, which may be subject of future research.
> 3.  **Comparative Analysis (Commercial/Learning-Based):** Thank you for the suggestion to include more learning-based baselines. We have carefully reviewed the suggested literature and found that these methods are not suitable for direct comparison due to a fundamental mismatch in the problem scale they are designed to address.
>
>     Specifically, the strongly related MAPDP framework by Zong et al. (2022) is architecturally constrained to small-scale instances, as the method relies on a Transformer-based encoder that computes attention across all nodes in the problem. Same holds for  Luo et al. (2023). This results in a computational complexity that scales quadratically (O(N²)) with the number of nodes, which is computationally intractable for the 50,000-node problems that are the focus of our work. Moreover, we did not find any code, and note that RL results strongly depend on the implementation.
>
>     Similarly, other works are designed for other settings, e.g., Li et al. (2021), Qiu et al. (2022) and Sun et al. (2023) are designed for single-vehicle settings. Finally, Hou et al. (2023) did not provide code. Our paper's primary contribution is a new paradigm for extreme-scale PDP, a domain where prior learning methods cannot operate. For this reason, we focus our comparison on these scalable, open-source baselines instead of commercial tools like Hexaly. We will clarify this positioning in our related work section.
>
> We thank the reviewer for their critical and constructive comments.

---

### Review · Reviewer_5UMp · 2025-07-28

**Summary Of Contributions:**

This paper proposes a novel method for the (capacitated) Vehicular Routing Problem (VRP) by first showcasing the mean-field limiting solution and then discretizing it back down into the combinatorial domain. They showcase state-of-the-art performance compared to existing baselines.

**Audience:**

Yes

**Audience Explanation:**

In general the application of mean-field limit approaches for combinatorial problems is interesting for large scale optimization.

**Broader Impact Concerns:**

No broader impact concerns

**Claims And Evidence:**

Yes

**Claims Explanation:**

The work is well motivated by first looking at the distributional mean-field limit and then “projecting” onto a finite space. The work is generally clear and the figures are appropriate for showing the general idea.

Looking at equation 2-3: Is this connected to an optimal transport (Wasserstein) flow? This somewhat looks like a continuity equation (2) and a transport map (3). There is probably some connection since your method ultimately estimates a pushforward (3) and a continuity term: ignoring positivity: $\nu_{t+1}(s,d) - \nu_t(s,d) = - \mu_t(s) T_t(s,d)$. If we don’t have mass-flow we get $\partial_t\nu_t(s,d) = -\mu_t(s) T_t(s,d)$, i.e. continuity with a sink $-\mu_t(s)T_t(s,d)$. As such I think it would be appropriate to cite existing work on OT+VRP like Santambrogio (https://arxiv.org/abs/1009.3857) or Huang (https://arxiv.org/abs/2012.08388), both of which use essentially the dynamic Benamou-Brenier OT formulation.

This is more of a remark, but I don’t quite understand the utility of the theorems/corollaries (2.1, 2.2, 3.1): Usually propagation of chaos is used to establish a link between a finite nonparametric Mean-Field estimate (i.e. approximating p(x) via N bins) and the true distribution p(x). This means that N is a free parameter one can alter to improve the approximation: If I want a certain $\varepsilon-\delta$ bound I can set $N$ high enough. However, here the N corresponds to the number of vehicles/objects, so it is not a free parameter. If I’m given a specific realization of a VRP, then the $N$ is fixed, so in some sense the guarantee should  be the other way around (given $N$, you prove $\varepsilon,\delta$). From everything I know about MFC, the $N$ one might need for decent approximation is huge.

A strength of this paper is the good delineation of the paper’s limitations: Capacities>1 is exponential, non-pickup-delivery VRPs are not really modellable.
I think you should strengthen your explanation of MFVRP-F vs MFVRP-FD: I think the distinction is that in MFVRP-F you just update the distributions themselves, while in MFVRP-FD you sample the discrete actions and update the distributions with those, but this is not entirely clear.
In Algorithm 2, line 5: Should this reference Algorithm 1 rather than Algorithm 2?
Regarding the results in table 1: Is this one policy (pre-)trained for all settings (uniform, cities, finite) or one policy per setting (i.e. one for uniform, one for cities, one for finite)?
This is somewhat important since if every setting needs one model, the pretraining time might dominate in the real world. In fact, an interesting benchmark to consider is what happens if one model was trained on e.g. uniform (and only uniform) and evaluated on e.g. cities. This would showcase on what level optimization performance hinges on pretraining performance.

What happens if K is larger than 5? Does the method still work? Having just 3-5 clusters is really limiting in practice.

Why is c_miss and r_bonus needed? Shouldn’t the choice of c_d remove the local optima?

On page 8 “For learning a general solution under fixed K, we dynamically generate every possible scenario by sampling $\nu_0$ and cluster centroids randomly according to flat Dirichlet and uniform distributions respectively”: should this be “any” scenario? I doubt that you can generate every scenario.

**Requested Changes:**

Test larger problems beyond K=5

Clarify the connection of this to existing classical OT-VRP approaches (see above).

Clarify the utility of theorems/corollaries (2.1, 2.2, 3.1).

---

> ### Author Response · Authors · 2025-08-17
>
> We are grateful for your positive and insightful review. We address each point below.
>
> **Concerns/Questions:**
>
> 1.  **Connection to Optimal Transport (OT):** We deeply appreciate the powerful theoretical lens. The reviewer is correct; we optimize for a discrete conditional optimal transport flow from depot back to itself, and have two coupled flows, which differentiates our problem from standard OT. The condition for the flow is to fulfill a total minimum flow over all timesteps at each location via the continuity equation with sink -- to transport all the objects -- at minimal cost. This is a very interesting connection and connects OT with VRP via mean-fields. We revised the paper to include this discussion in Section 2.3 and included the indicated references.
> 2.  **Utility of theorems/corollaries (2.1, 2.2, 3.1) - N is fixed:** Thank you for the question! Indeed, in MFC it is usually required to achieve uniform, usually quantified convergence for global optimality, and while $\mathcal X$ is finite, we have compact $\mathcal P(\mathcal X)$ and automatic uniform convergence. But in our case, we have revised the proofs to include results beyond asymptotic convergence to show that the error actually scales with $\mathcal O(1 / \sqrt N)$, and exact bounds for $\epsilon$ given $N$ could be computed inductively as in the proof. Moreover, for the corollaries, we changed our result to PAC-style guarantees using these quantified convergence rates, i.e. $\epsilon$-estimates given $N$ for any fixed confidence $1-\delta$. In practice, one may then insert their desired confidence level $\delta$ and obtain probable suboptimality bounds for their $N$-agent VRP, which makes the theoretical results useful in practice.
> 3.  **MFVRP-F vs MFVRP-FD distinction:** The reviewer's understanding is correct. MFVRP-F operates on continuous mean-field distributions, while MFVRP-FD samples discrete actions for individual vehicles. We strengthened this explanation in Section 3.2.
> 4.  **Algorithm 2, line 5:** We apologize for this typo, the reviewer is correct.
> 5.  **Table 1 policy training:** MFVRP-G is indeed already a single policy trained to generalize across diverse initial distributions for a *fixed* K. MFVRP-F fine-tunes this on a *specific* VRP instance. We will clarify this in the experimental details.
> 6.  **K > 5:** We acknowledge that increasing K significantly increases the action space, making the RL problem harder. This is a key limitation and a major area for future research, as noted in our conclusion. To illustrate this, we added a brief experiment in the appendix for K=6,7,8 which demonstrates the scalability challenge posed by the K^2-dimensional action space. High-dimensional continuous-action RL still remains very challenging.
> 7.  **$c_{miss}$ and $r_{bonus}$:** $r_{bonus}$ is needed, because agents are also forced to move back to the depot at the end of an episode, which we will clarify. $c_{miss}$ was indeed introduced only as reward shaping for RL, and could be left out.
> 8.  **"every possible scenario" phrasing:** We meant every possible limiting MFVRP scenario for fixed $K$, ignoring corresponding finite problems, since we sample all possible distributions of start and end locations + the locations of clusters. We rephrased accordingly. Any cluster locations + distributions over them will eventually be sampled (it suffices to sample up to a small difference, as we have continuous variables and continuity of the problem in them).
>
> We are thankful for the reviewer's detailed and constructive comments.

---

> > ### Comment · Reviewer_5UMp · 2025-08-20
> >
> > Thank you for your comment.
> >
> > I had a look at your additional scaling results: There seems to be a significant limitation when scaling K which is fine: every work has limitations (Please check your figure labels: A.1 and A.2 both point to figure 5 not figure 5 and 6). In general, this framework seems to be applicable for large scale, but simplistic VRPs (unit capacity, so no multi-pickup, low K, but large numbers of nodes) which may be useful in some circumstances.

---

### Review · Reviewer_vDEa · 2025-08-05

**Summary Of Contributions:**

This paper seeks to tackle very large scale unit-capacity pickup-and-delivery routing by casting the many-vehicle system into a mean-field formulation. The idea is to track distributions of trucks and package source-destination pairs instead of tracking every truck and package, thus turning a multi-agent problem into a single-agent MDP whose action is a transport map over a clustered discretization of space. The proposed approach then applies RL on this mean-field MDP. The paper also provides asymptotic guarantees and a discretization error bound from clustering. Overall, I find this work interesting, and the results seem quite convincing, demonstrating that the proposed mean-field approach is much faster than OR-Tools/PyVRP at very large scales. I find the work interesting and promising, but I have several reservations (evaluation scope, baseline coverage, and some clarity issues) which I believe should be addressed before publication.

**Additional Comments:**

Minor comments:

1. For example, (Zhou et al., 2024) considers -> For example, (Zhou et al., 2024) consider. This grammatical error is throughout the paper, please correct all the instances.
2. Proofs for theoretical results may be found in the Appendix- > can be found
3. Examplary system with four states -> Exemplary
4. As a finite problem of interest, one may consider vehicle routing problems with pickup and delivery -> this sentence reads super awkward to me and is not clear what the authors meant.

**Audience:**

Yes

**Audience Explanation:**

Overall, I find this paper's idea and results interesting and I believe after a revision (there are some inconsistencies and discrepancies) it can contribute to the scientific community in general and to TMLR's audience in particular.

**Claims And Evidence:**

Yes

**Claims Explanation:**

Most of the claims seem correct with few exceptions as detailed in the other parts of my review.

**Requested Changes:**

I have several questions, concerns and suggestions as elaborated below.

Concerns:

1. First and foremost, the title mentions VRP, but the studied problem is not the classic VRP. I would suggest the authors to revise the title to correctly reflect this paper's scope and contributions. For example, something along the lines " A Mean-Field RL for Large-Scale Unit-Capacity Pickup-and-Delivery Problems."

2. In section 2.2 it is stated that the constant c_d > 2/M diam(X), but in the following section the authors state that they chose c to be 2 diam(X), is this a typo or an oversight ?

3. In section 1.2, it is stated that several prior works allow sub-minute training times, which is also met in this paper. However, the reported results in section 4.1 state that fine-tuning takes around 3 minutes per instance and pre-training takes roughly 1 hour per K. This should be clarified and the language should be adjusted.

4. The scope of the evaluation is limited to synthetic data generated by the authors. In the "Related Works" section, several PDP papers are cited. Incorporating the datasets employed/generated in those works (or public PDP datasets) into the current evaluation not only would provide external validity to this work but will also allow for direct apples-to-apples comparisons with prior works. Furthermore, this will also facilitate future works' comparison with the results reported in this paper.

5. On a related note, the current set of benchmarks omits learning-based approaches from prior work. This weakens the empirical positioning of the contribution: it remains unclear how the proposed method compares to existing neural/learning-based VRP solvers reviewed in the Related Work section. Please either (i) include representative learning baselines, or (ii) provide a clear rationale for their exclusion, and discuss how those factors affect comparability.

Questions:

1. T_t grows quadratically with K. Have the authors considered any methods to reduce action dimension (e.g., low rank factorization ) ?

2. Have the authors considered post-processing steps other than delivering leftovers randomly ?

Suggestions:

1. The paper’s structure, especially the Introduction, can be made clearer. I recommend (i) stating the precise problem variant early (unit-capacity PDP, no time windows, tour-length cap), (ii) briefly explaining the link to CVRP used for baselines, and then (iii) introducing the mean-field perspective and contributions. This would anchor the reader before the methodological details and make the scope unambiguous.

---

> ### Author Response · Authors · 2025-08-17
>
> We appreciate your thorough review and positive assessment of our work. We address the raised concerns and suggestions below.
>
> **Concerns:**
>
> 1.  **Title:** We will revise the title to more accurately reflect the scope and contributions, adopting the title "Mean-Field RL for Large-Scale Unit-Capacity Pickup-and-Delivery Problems."
> 2.  **Section 2.2 constant $c_d$:** We will correct the inconsistency in Section 2.2. The condition $c_d \geq 2 \mathrm{diam}(X)$ is appropriate for the mean-field formulation to de-incentivize non-delivery.
> 3.  **Section 1.2 training times:** We will clarify to distinguish the actual training duration correctly (minutes for fine-tuning, hours for pre-training).
> 4.  **Evaluation scope (synthetic data):** We acknowledge the value of public PDP datasets. Our focus was on demonstrating scalability to very large instances (up to 50,000 nodes), which are not typically covered by existing standard public PDP datasets.
> 5.  **Baseline coverage (learning-based approaches):** Thank you for the suggestion to include more learning-based baselines. We have carefully reviewed the literature and found that these methods are not suitable for direct comparison due to a fundamental mismatch in the problem scale they are designed to address.
>
>     Specifically, the strongly related MAPDP framework by Zong et al. (2022) is architecturally constrained to small-scale instances, as the method relies on a Transformer-based encoder that computes attention across all nodes in the problem. Same holds for  Luo et al. (2023). This results in a computational complexity that scales quadratically (O(N²)) with the number of nodes, which is computationally intractable for the 50,000-node problems that are the focus of our work. Moreover, we did not find any code, and note that RL results strongly depend on the implementation.
>
>     Similarly, other works are designed for other settings, e.g., Li et al. (2021), Qiu et al. (2022) and Sun et al. (2023) are designed for single-vehicle settings. Finally, Hou et al. (2023) did not provide code. Our paper's primary contribution is a new paradigm for extreme-scale PDP, a domain where prior learning methods cannot operate. We will clarify this positioning in our related work section.
>
> **Questions:**
>
> 1.  **$T_t$ quadratic growth with $K$:** We acknowledge this limitation and mention it as future work. Methods like low-rank factorization are promising for future exploration. We tried training rank-1 approximations for the transition matrix, which reduces complexity from $K^2$ to $K$, and the curves were added to the appendix, but the results are significantly worse.
> 2.  **Post-processing steps:** In principle, one can apply any other heuristic to transport the remainder. However, for large scales, the fraction of undelivered objects is theoretically guaranteed to be small, leading us to favor simplicity here.
>
> **Suggestions:**
>
> 1.  **Paper structure/clarity:** We restructured the introduction to clearly state the problem variant, its link to CVRP, and then introduce the mean-field perspective and contributions for improved readability.
>
>
> **Minor comments:**
>
> We changed "may be found" to "are found," correct the text around references, "Examplary" to "Exemplary," and rephrased the sentence in Section 2.1.
>
> We thank the reviewer again for their constructive feedback.

---

> > ### Comment · Reviewer_vDEa · 2025-08-23
> >
> > I thank the authors for their reply, and appreciate the efforts put on the revision. The concerns regarding the evaluation scope and benchmarks, which were also brought up by other Reviewers, remain somewhat not fully addressed. However, reasonable justifications were provided, which are acceptable to me. In the response, the authors mention difficulties finding the code/datasets of prior studies. To break this cycle, I would like to kindly request the authors to open-source their entire codebase and the generated datasets. This will facilitate further research in this direction as well as improve the reproducibility of the results.

---

> > > ### Author Response · Authors · 2025-08-24
> > >
> > > We thank the reviewer for their follow-up and for their understanding. We agree that open-sourcing is crucial for reproducibility and for fostering future research in this direction, and are happy to confirm that we will make our codebase and generated datasets publicly available upon publication.

---

### Decision · Action_Editor_irUZ · 2025-08-26

**Recommendation:** Accept with minor revision

**Additional Comments:**

The reviewers found this work interesting, well motivated and promising, the proposed MFC formulation for VRPs is novel, and the results are convincing with good performance on large-scale problems. Many concerns have been raised on the limitations of the proposed method (e.g., unit capacity, scaling exponentially in vehicle capacity, scaling poorly with cluster size), clarifications, comparison with other learning-based algorithms, and selected benchmark problems. The authors have provided concise responses to address these concerns and revised the manuscript accordingly.

After rebuttal, this work received mixed ratings (1 accept, 1 weak accept, and 1 reject). Reviewer vDEa believes the concerns regarding the evaluation scope and benchmarks (also raised by other reviewers) are not fully addressed but the provided justifications are reasonable, therefore votes to accept this work. Reviewer 5UMp thinks the proposed framework could be applicable for large-scale but simplistic VRPs (limited by unit capacity, scaling exponentially in vehicle capacity, scaling poorly with cluster size), and appreciates this paper's clear discussion of limitations. This reviewer leans toward accepting this work. In the official recommendation, reviewer NtR6 believes the concerns on strong assumption (unit capacity), comparison to state-of-the-art algorithms, and evaluation are not well addressed, and hence votes to reject this work.

I read this paper in detail and agree with all reviewers that the proposed MFC formulation for routing problems is novel and interesting, and some individuals, especially those working on learning-based combinatorial optimization, could be interested in the findings of this paper. I also fully agree with the reviewers that the current work still has limitations on the proposed method's ability to solve complex real-world problems, and the experimental study can be further improved. Given that the method's limitations have been properly discussed in the revised paper and a reasonable justification has been provided to support the experimental setting, I believe the current claims made in this work are supported by clear evidence, and therefore recommend **accepting this work, but with a "minor" (somewhat like a major) revision**.

To further address the concerns raised by the reviewers and improve the quality of this work, I have the following suggestions.

**Unit Capacity:** Please explicitly discuss the proposed method's limitation on unit capacity in the limitation subsection of Section 2.5 and Section 5 (in addition to the discussion on generality). If applicable, a concrete example on how to generalize the proposed method to tackle higher-than-unit capacity problems could be appreciated.

**Comparison with Other Learning-based Methods:** The authors' response claims that "[the other learning-based methods are] computationally intractable for the 50,000-node problems that are the focus of our work. Moreover, we did not find any code...". However, to my understanding, the current learning-based routing solvers with a heavy decoder approach, such as BQ-NCO[1], LEHD [2] (Luo et al. 2023 mentioned in the response), and SIGD [3], can generalize well to tackle large-scale routing problems even only trained on small-scale instances with 100 nodes. For example, please see the results on CVRP100K in SIL [4]. In addition, to my best knowledge, the codes and trained models of these methods are all fully open-sourced, and the results can be easily reproduced. Therefore, the current discussion on other learning-based methods is not accurate.

On the other hand, although SIL[4] can tackle large-scale CVRP instances, I fully understand that some adaptations could be needed to solve VRPPD. A specific inference and solution generation strategy could be needed to directly handle instances with the pickup-and-delivery attribute. In addition, as discussed in this paper, the unit-capacity pickup-and-delivery can be reformulated as a CVRP with asymmetric non-Euclidean distance. To use the current learning-based routing solvers to tackle the problem in this way, a distance-matrix-based encoder, such as those in [5,6], could be needed.

In summary, although a direct comparison is still not very straightforward, at least a thorough and accurate discussion with other learning-based methods is needed.

[1] BQ-NCO: Bisimulation Quotienting for Efficient Neural Combinatorial Optimization. NeurIPS 2023.

[2] Neural Combinatorial Optimization with Heavy Decoder: Toward Large Scale Generalization. NeurIPS 2023.

[3] Self-improvement for neural combinatorial optimization: Sample without replacement, but improvement. TMLR 2024.

[4] Boosting Neural Combinatorial Optimization for Large-Scale Vehicle Routing Problems. ICLR 2025.

[5] Matrix Encoding Networks for Neural Combinatorial Optimization. NeurIPS 2021.

[6] Neural Combinatorial Optimization for Real-World Routing. arXiv:2503.16159.


**Evaluations/Discussions on (Small-Scale) public PDP datasets:** I fully understand that the focus of this work is on scalability to large-scale instances. However, a clear discussion on its (not strong) results on small to medium scale instances (e.g., 100 - 5k nodes as reported in this paper) is still very helpful to fully understand the pros and cons of the proposed methods. I also agree with the reviewers that reporting results on standard public PDP datasets will also allow for direct apples-to-apples comparisons with prior works.

A key research question to answer: Why does MFVRP not perform so well on small to medium scale instances, even with finetuning?

**Performance of PyVRP:** If I understand correctly, the method used in the PyVRP package is the HGS algorithm, which can achieve reasonably good performance on CVRP50K instances. Why are the HGS results on M = 50,000 missing in Table 2?

In addition, the computational budget of HGS can be easily adjusted, which means it can still produce a result with a short runtime (but maybe with low quality). Therefore, the claim that "PyVRP takes very long to produce results at very large scales" needs more explanation.

In the camera-ready version, I expect the authors to properly address the concerns raised above. In addition, please ensure that all discussions are carefully incorporated into the paper and release the codebase and generated datasets as promised.

**Audience:**

Yes

**Audience Explanation:**

This work proposes a novel learning-based mean-field control approach to tackle large-scale routing problems with unit-capacity pickup-and-delivery. All reviewers believe some individuals in TMLR's audience could be interested in the findings of this paper.

**Claims And Evidence:**

Yes

**Claims Explanation:**

This work proposes MFVRP, a reinforcement learning based mean-field control (MFC) method, for solving large-scale unit-capacity pickup-and-delivery problems. The key idea is to formulate the original discrete routing problem into an MFC problem, and then propose an efficient single-agent RL-based MFVRP method to tackle the MFC problem efficiently. Theoretical analysis is provided to support the proposed MFC-based method. Experimental results show the proposed MFVRP method can outperform some classic heuristic solvers on large-scale unit-capacity pickup-and-delivery problems with up to 50K nodes.

Two reviewers believe the claims made in this paper are supported by convincing and clear evidence, while one reviewer thinks the claims are not well-supported. Please refer to the comment below for more details.

---

> ### Author Response · Authors · 2025-09-08
>
> Thank you for your detailed feedback and accepting the paper. We have submitted the deanonymized version of the paper and have carefully addressed all your suggestions, updating the manuscript accordingly. For convenience, here are the changes:
>
> 1.  **Unit Capacity:** As suggested, we have explicitly discussed the unit-capacity limitation in the revised "Generality and Limitations" section (Sec 2.5). We now detail concretely how the framework could be extended to higher capacities by giving the example of capacity equal two, and note the resulting quadratic complexity.
>
> 2.  **Comparison with Other Learning-based Methods:** We have updated the "Related Work" section with a more accurate discussion. We now acknowledge that methods like LEHD and SIGD generalize to large scales but clarify that a direct benchmark is non-trivial. Applying them to our specific PDP setting would require significant adaptations (e.g., your proposed distance-matrix encoders for the asymmetric costs etc.), which we leave for future work.
>
> 3.  **Evaluation on Small-Scale Instances:** We have added a discussion directly addressing why MFVRP is less competitive on smaller problems. The performance gap stems partially from the inherent approximation error of the mean-field and clustering approach, which is more significant for smaller N. Traditional methods operating on the exact problem graph are naturally more effective at that scale. We also clarified that our focus on extreme scalability (50k nodes) is why we generated new datasets, as standard PDP benchmarks are typically much smaller.
>
>     Moreover, we also attribute suboptimality to the limited optimality of the RL solution. In particular, we note that the RL solution in Figure 3 is slightly unstable, i.e. the solution deteriorates at a certain point and does not stay in a local optimum. Further work invested into reward shaping, hyperparameter tuning and neural network architecture may still improve results. This is supported also by the limited RL performance for higher $K$ as seen in the Appendix.
>
>     The discussions were added to the experiments section, and two additional small experiments for the above arguments were added to the Appendix.
>
> 4.  **Performance of PyVRP:** We have clarified in the results section why PyVRP (HGS) results are missing for M=50,000. It failed to produce a solution within our time limit (>120 mins), primarily due to the pre-processing bottleneck of constructing the quadratic distance matrix sequentially using Python loops, since the edges are internally kept as a Python list. This becomes intractable at our scale and is unfortunately not parallelized on GPUs. Meanwhile, we note the mean field solution has a linear complexity, which may also eventually outscale standard parallelized attention-based transformers on GPUs. Here is the problematic code snippet: https://github.com/PyVRP/PyVRP/blob/main/pyvrp/Model.py#L81
>
> We have incorporated these discussions carefully and reaffirmed our commitment to open source by having included a link to our codebase with data generation scripts. Thank you again for your valuable guidance!